# Decomposed Direct Preference Optimization for Structure-Based Drug Design

**Xiwei Cheng**[*]                                                 *cheng.xiw@northeastern.edu*
*Khoury College of Computer Sciences, Northeastern University*

**Xiangxin Zhou**[*]                                               *zhouxiangxin1998@gmail.com*
*ByteDance Seed*
*School of Artificial Intelligence, University of Chinese Academy of Sciences*
*New Laboratory of Pattern Recognition (NLPR), State Key Laboratory of Multimodal Artificial Intelligence Systems*
*(MAIS), Institute of Automation, Chinese Academy of Sciences (CASIA)*

**Yu Bao**                                                          *baoyu.001@bytedance.com*
*ByteDance Seed*

**Yuwei Yang**                                                      *yuwei.yang@bytedance.com*
*ByteDance*

**Quanquan Gu**                                                     *quanquan.gu@bytedance.com*
*ByteDance Seed*

**Reviewed on OpenReview:** *https://openreview.net/forum?id=dwSpo5DRk8*

## Abstract

Diffusion models have achieved promising results for Structure-Based Drug Design (SBDD). Nevertheless, high-quality protein subpockets and ligand data are relatively scarce, which hinders the models' generation capabilities. Recently, Direct Preference Optimization (DPO) has emerged as a pivotal tool for aligning generative models with human preferences. In this paper, we propose DECOMPDPO, a structure-based optimization method that aligns diffusion models with pharmaceutical needs using multi-granularity preference pairs. DECOMPDPO introduces decomposition into the optimization objectives and obtains preference pairs at the molecule or decomposed substructure level based on each objective's decomposability. Additionally, DECOMPDPO introduces a physics-informed energy term to ensure reasonable molecular conformations in the optimization results. Notably, DECOMPDPO can be effectively used for two main purposes: (1) fine-tuning pretrained diffusion models for molecule generation across various protein families, and (2) molecular optimization given a specific protein subpocket after generation. Extensive experiments on the CrossDocked2020 benchmark show that DECOMPDPO significantly improves model performance, achieving up to 98.5% Med. High Affinity and a 43.9% success rate for molecule generation, and 100% Med. High Affinity and a 52.1% success rate for targeted molecule optimization. Code is available at https://github.com/laviaf/DecompDPO.

## 1 Introduction

Structure-based drug design (SBDD) (Anderson, 2003) is a strategic approach in medicinal chemistry and pharmaceutical research that utilizes 3D structures of biomolecules to guide the design and optimization of new therapeutic agents. The goal of SBDD is to design molecules that bind to specific protein targets. Recent studies viewed this task as a data-driven conditional generative problem, introducing powerful generative

---

[*]Equal contribution (this work was done during Xiwei and Xiangxin's internship at ByteDance).

models with geometric deep learning (Powers et al., 2023). For example, Peng et al. (2022); Zhang & Liu (2023) proposed generating atoms or fragments sequentially by an SE(3)-equivariant auto-regressive model, Luo et al. (2021); Peng et al. (2022); Guan et al. (2023a) introduced diffusion models (Ho et al., 2020) to model distributions over ligand atom types and positions.

A significant bottleneck for the development of generative models in SBDD is the scarcity of high-quality protein-ligand complex data (Vamathevan et al., 2019). While large-scale datasets have spurred rapid advances in fields such as natural language processing, the collection of protein-ligand binding data is notably more challenging and limited due to the complex and resource-intensive experimental procedures. Notably, the CrossDocked2020 dataset (Francoeur et al., 2020), a widely-used dataset for SBDD, augments existing data by docking ligands into similar protein pockets in the Protein Data Bank. Although this increases dataset size, it may unavoidably introduce some low-quality data. As highlighted by Zhou et al. (2024a), the ligands in the CrossDocked2020 dataset have moderate binding affinities, which do not meet the stringent demands of drug design. Moreover, the number of unique ligands remains the same before and after this data augmentation, limiting generative models from learning diverse and high-quality molecules.

To address the aforementioned challenge, Xie et al. (2021); Fu et al. (2022) provided a straightforward method for searching molecules with desired properties in the extensive chemical space. However, pure searching or optimization methods lack generative capabilities and fall short in the diversity of the designed molecules. Zhou et al. (2024a) integrated a conditional diffusion model with optimization by iteratively selecting better molecular substructures as generation conditions. This method achieves better properties while maintaining a certain level of diversity. Nonetheless, the performance of this method is still limited due to fixed model parameters during the optimization process.

To break the bottleneck, we introduce DECOMPDPO, a multi-objective optimization framework that aligns diffusion models with practical pharmaceutical requirements using generated data. Inspired by the decomposition nature of ligand molecules, DECOMPDPO introduces decomposition into the optimization objective to provide more flexibility in preference selection and alignment. Based on each objective's decomposability, DECOMPDPO directly aligns the model with full-molecule level preferences using GLOBALDPO or LOCALDPO with decomposed-substructure level preferences. Recognizing the importance of physically realistic molecule conformations in drug discovery, DECOMPDPO integrates physics-informed energy terms to penalize molecules with poor conformations. Additionally, a linear beta schedule is proposed for improving optimization efficiency. We demonstrate the effectiveness of DECOMPDPO in two scenarios: structure-based molecule generation and molecule optimization, showing that DECOMPDPO significantly outperforms existing baselines. We highlight our contributions as follows:

- We propose DECOMPDPO, which introduces decomposition into the optimization objectives for more effective and flexible alignment of diffusion-based generative models with real-world pharmaceutical requirements using multi-granularity preferences.

- Our approach is applicable to both structure-based molecule generation and optimization. Notably, DECOMPDPO achieves 98.5% Med. High Affinity and a 43.9% success rate for molecule generation, and 100% Med. High Affinity and a 52.1% success rate for molecule optimization on CrossDocked2020 dataset.

- To the best of our knowledge, we are one of the first works to introduce preference alignment to structure-based drug design. Our approach aligns the generative models for SBDD with the practical requirements of drug discovery.

Recently, an independent concurrent work by Gu et al. (2024) also employs preference alignment for fine-tuning diffusion models in SBDD. Specifically, they regularized the DPO objective to mitigate overfitting on winning data. However, they primarily focused on binding affinity optimization and do not check the sanity and conformational soundness of generated molecules, which are essential in practical drug design. Compared to Gu et al. (2024), we construct preference pairs at both the full-molecule and decomposed-substructure levels for enhanced optimization performance and flexibility, and integrate physics-informed energy terms to penalize unreasonable molecular conformations. Beyond binding affinity optimization, the effectiveness of DECOMPDPO is demonstrated under a broader multi-objective setting to generate molecules better aligned with pharmaceutical needs, with further evidence of its strong performance through iterative fine-tuning for molecule optimization.

## 2 Related Work

**Structure-based Drug Design**  Structure-based drug design (SBDD) aims to design ligand molecules that can bind to specific protein targets. Recent efforts have been made to enhance the efficiency of modeling molecule distributions. Ragoza et al. (2022) employed variational autoencoder to generate 3D molecules in atomic density grids. Autoregressive approaches (Luo et al., 2021; Peng et al., 2022; Liu et al., 2022) generate 3D molecules atom by atom, while Zhang et al. (2022) extends to predict molecular fragments in an auto-regressive way. Recently, Guan et al. (2023a); Schneuing et al. (2022); Lin et al. (2022) introduced diffusion models to SBDD. Building upon these developments, some recent studies have sought to further enhance SBDD methods by incorporating biochemical prior knowledge. DecompDiff (Guan et al., 2023b) decomposed ligands into substructures and generated molecules with decomposed priors and validity guidance in diffusion. DrugGPS (Zhang & Liu, 2023) incorporated subpocket similarities to augment molecule generation. IPDiff (Huang et al., 2023) addressed the inconsistency between forward and reverse diffusion by leveraging a pre-trained protein-ligand interaction prior. Despite these advancements, effectively generating molecules that meet real-world drug development criteria remains challenging due to the fundamental mismatch between distribution learning and reward-guided generation. In this work, we aim to bridge this inconsistency by aligning diffusion model distributions directly with real-world pharmaceutical objectives, enhancing the molecule generation efficiency.

**Structure-based Molecule Optimization**  In addition to simply generative modeling of existing protein-ligand pairs, some researchers leveraged optimization algorithms to design molecules with desired properties. AutoGrow 4 (Spiegel & Durrant, 2020), RGA (Fu et al., 2022), and DecompOpt (Zhou et al., 2024a) take a specific protein target as input and perform optimization using oracle functions to evaluate the fitness of generated molecules. Specifically, RGA (Fu et al., 2022) used neural networks to guide the genetic algorithm with reinforcement learning. DecompOpt (Zhou et al., 2024a) employed a controllable diffusion model conditioned on protein subpockets and substructures for iterative optimization. In addition, Reidenbach (2024); Shen et al. (2023) performed structure-based optimization in lower dimensions, such as latent vectors or 2D graphs. Beyond optimizing towards a single protein target, methods such as PILOT (Cremer et al., 2024) reweighted diffusion trajectories toward particular objectives using importance sampling, KGDiff (Qian et al., 2024) guided diffusion trajectories with gradients from an additional expert network, and TAGMol (Dorna et al., 2024) guided continuous coordinates in both the training and sampling processes.

A complementary line of work optimizes ligand-only 2D graphs or SMILES strings without explicit 3D protein conditioning during generation. Reinvent (Olivecrona et al., 2017) uses reinforcement learning to fine-tune SMILES generators with property-specific reward signals. GraphGA (Jensen, 2019) employs a genetic algorithm over molecular graphs guided by reward functions. MARS (Xie et al., 2021) adopts Monte Carlo tree search for fragment-based graph optimization. GEAM (Lee et al., 2023b) learns a task-specific fragment vocabulary for reward-guided molecular assembly, while Saturn (Guo & Schwaller, 2024) combines Mamba models with memory-based experience replay for efficient optimization. These methods have shown effectiveness in property-driven optimization tasks but operate without explicit 3D protein information, where spatial complementarity and interaction geometry are essential.

In this work, we target SBDD and therefore benchmark against SBDD baselines to ensure a fair comparison in inputs, hypothesis space, and evaluation protocol. Our model directly aligns the diffusion model with preferences, and is applicable to both target-specific and general-purpose SBDD optimization settings.

**Learning from Human/AI Feedback**  Likelihood-based training alone can fail to meet user-defined preferences for generative models. Reinforcement learning from human or AI feedback (Ziegler et al., 2019; Stiennon et al., 2020; Ouyang et al., 2022; Lee et al., 2023a; Bai et al., 2022) addressed this by first learning a reward model from data annotated by human or AI, then fine-tuning the generative model through policy-gradient methods (Christiano et al., 2017; Schulman et al., 2017). Similar techniques have also been introduced to diffusion models for text-to-image generation (Black et al., 2023; Fan et al., 2024; Zhang et al., 2024). Recently, Direct Preference Optimization (DPO) (Rafailov et al., 2024) further simplified the process by directly optimizing on preference pairs without an explicit reward model. Wallace et al. (2023) re-formulated DPO and derived Diffusion-DPO for aligning text-to-image diffusion models. While these methods primarily

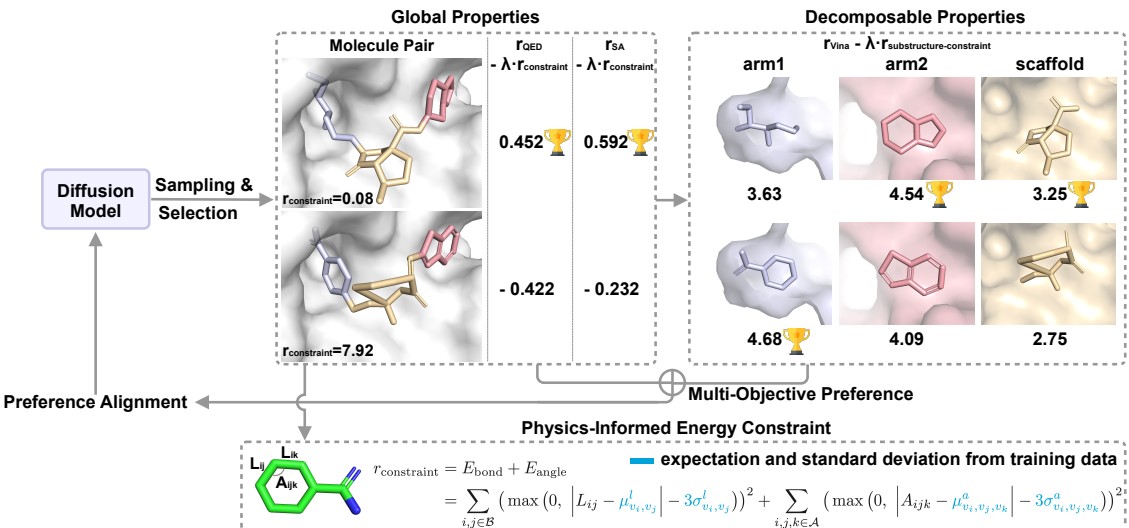

Figure 1: Overview of DECOMPDPO. (a) Sample molecules and select molecule pairs for each target protein using a pre-trained diffusion model; (b) Construct physically constrained preference for each optimization objective based on its decomposability; (c) Compute the DECOMPDPO loss and align the diffusion model with the multi-objective preference.

focus on natural language or image generation, Zhou et al. (2024b) proposed to fine-tune diffusion models for antibody design by DPO targeting low Rosetta energy. In our work, we introduce preference alignment to improve the desired properties of generated molecules and propose specialized methods to improve the performance of DPO in the scenario of SBDD.

## 3 Method

In this section, we introduce DECOMPDPO, which aligns diffusion models with pharmaceutical needs using physically constrained multi-granularity preferences (Figure 1). We first define the SBDD task and introduce the decomposed diffusion model in Section 3.1. We then incorporate decomposition into the optimization objectives for multi-objective preference alignment in Section 3.2. Recognizing the importance of maintaining physically realistic molecular conformations during optimization, we introduce physics-informed energy terms for penalizing rewards in Section 3.3. Finally, we propose a linear beta schedule to improve the optimization efficiency for diffusion models (Section 3.4).

### 3.1 Preliminaries

In the context of SBDD, generative models aim to generate ligands $\mathcal{M} = \{(\boldsymbol{x}_i^{\mathcal{M}}, \boldsymbol{v}_i^{\mathcal{M}}, \boldsymbol{b}_{ij}^{\mathcal{M}})\}_{i,j \in \{1,\cdots,N_{\mathcal{M}}\}}$ that bind to a specific protein binding site, represented as $\mathcal{P} = \{(\boldsymbol{x}_i^{\mathcal{P}}, \boldsymbol{v}_i^{\mathcal{P}})\}_{i \in \{1,\cdots,N_{\mathcal{P}}\}}$. Here, $N_{\mathcal{P}}$ and $N_{\mathcal{M}}$ are the number of atoms in the protein and ligand, respectively; $\boldsymbol{x} \in \mathbb{R}^3, \boldsymbol{v} \in \mathbb{R}_a^K, \boldsymbol{b}_{ij} \in \mathbb{R}_b^K$ represents the 3D atom coordinates, the atom types, and the bonds between atoms, where $K_a$ and $K_b$ represent the number of atom and bond types.

Following the decomposed diffusion model introduced by Guan et al. (2023b), each ligand is decomposed into fragments $\mathcal{K}$, comprising several arms $\mathcal{A}$ connected by at most one scaffold $\mathcal{S}$ ($|\mathcal{A}| \geq 1, |\mathcal{S}| \leq 1, K = |\mathcal{K}| = |\mathcal{A}| + |\mathcal{S}|$). Based on the decomposed substructures, informative data-dependent priors $\mathbb{O}_{\mathcal{P}} = \{\boldsymbol{\mu}_{1:K}, \boldsymbol{\Sigma}_{1:K}, \mathbf{H}\}$ are estimated from atom positions by maximum likelihood estimation, where $\boldsymbol{\mu}_k \in \mathbb{R}^3$ represents the prior center, $\boldsymbol{\Sigma}_k \in \mathbb{R}^{3\times3}$ represents the prior covariance matrix, and $\mathbf{H} = \{\eta \in \{0,1\}^{N_M \times K} | \sum_{k=1}^K \eta_{ik} = 1\}$ represents the prior-atom mapping for $\mathcal{M}$. This data-dependent prior enhances the training efficiency of the diffusion model, where $\mathcal{M}$ is gradually diffused with a fixed schedule $\{\lambda_t\}_{t=1,\cdots,T}$. We denote $\alpha_t = 1 - \lambda_t$ and $\bar{\alpha}_t = \prod_{s=1}^t \alpha_t$. The $i$-th atom position is shifted to its corresponding prior center: $\tilde{\mathbf{x}}_t^i = \mathbf{x}_t^i - (\mathbf{H}^i)^\top \boldsymbol{\mu}$.

The noisy data distribution at time $t$ derived from the distribution at time $t-1$ is computed as follows:

$$q(\tilde{\mathbf{x}}_t|\tilde{\mathbf{x}}_{t-1}, \mathcal{P}) = \prod_{i=1}^{N_{\mathcal{M}}} \mathcal{N}(\tilde{\mathbf{x}}_t^i; \tilde{\mathbf{x}}_{t-1}^i, \lambda_t (\mathbf{H}^i)^\top \boldsymbol{\Sigma}),$$

$$q(\mathbf{v}_t|\mathbf{v}_{t-1}, \mathcal{P}) = \prod_{i=1}^{N_{\mathcal{M}}} \mathcal{C}(\mathbf{v}_t^i|(1-\lambda_t)\mathbf{v}_{t-1}^i + \lambda_t/K_a), \quad (1)$$

$$q(\mathbf{b}_t|\mathbf{b}_{t-1}, \mathcal{P}) = \prod_{i=1}^{N_{\mathcal{M}} \times N_{\mathcal{M}}} \mathcal{C}(\mathbf{b}_t^i|(1-\lambda_t)\mathbf{b}_{t-1}^i + \lambda_t/K_b).$$

The perturbed structure is then fed into the prediction model. The reconstruction loss at time $t$ can be derived from the KL divergence as follows:

$$L^{(x)} = \mathbb{E}_t\left[||\mathbf{x}_0 - \hat{\mathbf{x}}_0||^2\right], \ L^{(v)} = \mathbb{E}_t\left[\sum_{k=1}^{K_a} \boldsymbol{c}(\mathbf{v}_t, \mathbf{v}_0)_k \log \frac{\boldsymbol{c}(\mathbf{v}_t, \mathbf{v}_0)_k}{\boldsymbol{c}(\mathbf{v}_t, \hat{\mathbf{v}}_0)_k}\right], \ L^{(b)} = \mathbb{E}_t\left[\sum_{k=1}^{K_b} \boldsymbol{c}(\mathbf{b}_t, \mathbf{b}_0)_k \log \frac{\boldsymbol{c}(\mathbf{b}_t, \mathbf{b}_0)_k}{\boldsymbol{c}(\mathbf{b}_t, \hat{\mathbf{b}}_0)_k}\right], \ (2)$$

where $(\mathbf{x}_0, \mathbf{v}_0, \mathbf{b}_0)$, $(\mathbf{x}_t, \mathbf{v}_t, \mathbf{b}_t)$, $(\hat{\mathbf{x}}_0, \hat{\mathbf{v}}_0, \hat{\mathbf{b}}_0)$, represent true atoms positions, types, and bonds types at time 0, time $t$, and predicted atoms positions, types, and bonds types at time $t \sim U[0,T]$; $\boldsymbol{c}$ denotes mixed categorical distribution with weight $\bar{\alpha}_t$ and $1 - \bar{\alpha}_t$. The overall loss is $L = L^{(x)} + \gamma_v L^{(v)} + \gamma_b L^{(b)}$, with $\gamma_v, \gamma_b$ as weights of reconstruction loss of atom and bond type. We provide more details for the model architecture in Appendix B. To better illustrate decomposition, we show a decomposed molecule with the arms highlighted in Figure 2.

## 3.2 Direct Preference Optimization in Decomposed Space

**Decomposable Optimization Objectives** In real pharmaceutical applications, drug candidates should possess multiple desirable properties, yet such molecules are rare among all known drug-like molecules, making distribution learning less efficient in generating desired molecules. Although previous methods like DecompOpt (Zhou et al., 2024a) fully exploit the power of conditional diffusion models through iterative generation, the model's performance is limited by static parameters learned from offline data. Direct preference optimization offers a simple yet efficient way to align models with pairwise preference data. Inspired by the success of ligand decomposition in improving generative power (Guan et al., 2023b; Zhou et al., 2024a), we introduce decomposition into optimization objectives in DECOMPDPO for greater flexibility in preference selection and alignment.

An optimization objective is considered decomposable if the overall score of a molecule is proportional to the sum of substructure-level scores, which implies that a substructure with a higher score will lead the molecule to have a higher overall score. For example, *Vina Minimize Score* is largely based on pairwise atomic interactions, with each substructure contributing its own interactions with the protein target and negligible inter-substructure interactions, making it decomposable. As shown in Figure 2, we validated the proportional relationship of *Vina Minimize Score* in our dataset. However, objectives like *QED* and *SA* are non-decomposable, as their calculations involve non-linear operations. We provide more statistical evidence in Appendix C.

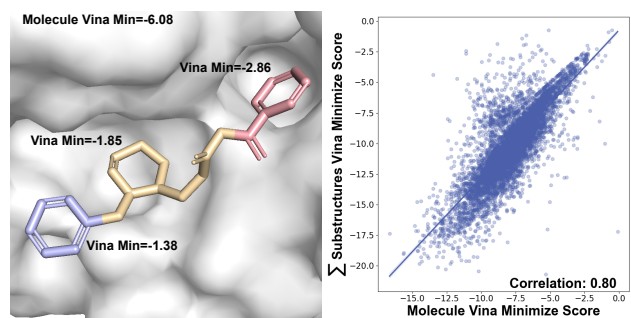

Figure 2: Illustration of decomposable objectives. Decompose a molecule into two arms (purple and pink) and a scaffold (yellow), where the sum of the substructures' Vina Minimize Scores equals to the molecule's (left). The Pearson correlation between molecule's and sum of substructure's Vina Minimize Scores in the training dataset (right).

**GlobalDPO** To align the model with practical pharmaceutical preferences, following RLHF (Ouyang et al., 2022), the pre-trained model is fine-tuned by maximizing certain reward functions with the

Kullback–Leibler (KL) divergence regularization:

$$\max_{p_\theta} \mathbb{E}_{\substack{\mathcal{P}\sim\mathcal{D},\\ \mathcal{M}\sim p_\theta(\mathcal{M}|\mathcal{P})}} \big[ r(\mathcal{M}, \mathcal{P}) \big] - \beta D_{\mathrm{KL}}\big( p_\theta(\mathcal{M} \mid \mathcal{P}) \,\|\, p_{\mathrm{ref}}(\mathcal{M} \mid \mathcal{P}) \big), \tag{3}$$

where $\beta > 0$ is a hyperparameter controlling the deviation from the reference model $p_{\mathrm{ref}}$. Recently, DPO (Rafailov et al., 2024) derives a pairwise training loss from equation 3, providing a simpler way to fine-tune the model with pairwise preference data:

$$\mathcal{L}_{\mathrm{DPO}} = -\mathbb{E}_{(\mathcal{P},\mathcal{M}^+,\mathcal{M}^-)\sim\mathcal{D}}\left[ \log\sigma\left( \beta\log\frac{p_\theta(\mathcal{M}^+ \mid \mathcal{P})}{p_{\mathrm{ref}}(\mathcal{M}^+ \mid \mathcal{P})} - \beta\log\frac{p_\theta(\mathcal{M}^- \mid \mathcal{P})}{p_{\mathrm{ref}}(\mathcal{M}^- \mid \mathcal{P})} \right) \right], \tag{4}$$

where $\mathcal{M}^+$ and $\mathcal{M}^-$ represent the preferred and less preferred molecules, respectively.

As $p_\theta(\mathcal{M}_0 \mid \mathcal{P})$ is intractable for diffusion models, following Diffusion-DPO (Wallace et al., 2023), we define the reward over the entire diffusion process $r(\mathcal{M}, \mathcal{P}) = \mathbb{E}_{p_\theta(\mathcal{M}_{1:T}|\mathcal{M}_0,\mathcal{P})}[R(\mathcal{M}_{1:T}, \mathcal{P})]$, where $\mathcal{M}_{1:T}$ denotes the diffusion trajectories from the reverse process $p_\theta$. By utilizing the evidence lower bound, the DPO loss is converted to:

$$\mathcal{L}_{\mathrm{Diffusion\text{-}DPO}} = -\mathbb{E}_{\substack{(\mathcal{P},\mathcal{M}^+,\mathcal{M}^-)\sim\mathcal{D}\\ \mathcal{M}_{1:T}^+\sim p_\theta(\mathcal{M}_{1:T}^+|\mathcal{M}_0^+,\mathcal{P})\\ \mathcal{M}_{1:T}^-\sim p_\theta(\mathcal{M}_{1:T}^-|\mathcal{M}_0^-,\mathcal{P})}}\left[ \log\sigma\left( \beta\mathbb{E}_{\mathcal{M}_{1:T}^+,\mathcal{M}_{0:T}^-}\left[ \log\frac{p_\theta(\mathcal{M}_{0:T}^+ \mid \mathcal{P})}{p_{\mathrm{ref}}(\mathcal{M}_{0:T}^+ \mid \mathcal{P})} - \log\frac{p_\theta(\mathcal{M}_{0:T}^- \mid \mathcal{P})}{p_{\mathrm{ref}}(\mathcal{M}_{1:T}^- \mid \mathcal{P})} \right] \right) \right]. \tag{5}$$

Following Wallace et al. (2023), we further approximate the intractable reverse probability $p_\theta$ with forward probability $q$ and use Jensen's inequality to externalize the expectation:

$$\mathcal{L}_{\mathrm{Diffusion\text{-}DPO}} = -\mathbb{E}_{\substack{(\mathcal{P},\mathcal{M}^+,\mathcal{M}^-)\sim\mathcal{D},t\sim\mathcal{U}(0,T),\\ \mathcal{M}_t^+\sim q(\mathcal{M}_t^+|\mathcal{M}_0^+),\\ \mathcal{M}_t^-\sim q(\mathcal{M}_t^-|\mathcal{M}_0^-)}}\left[ \log\sigma\left( \beta\left[ \log\frac{p_\theta(\mathcal{M}_{t-1}^+ \mid \mathcal{M}_t^+, \mathcal{P})}{p_{\mathrm{ref}}(\mathcal{M}_{t-1}^+ \mid \mathcal{M}_t^+, \mathcal{P})} - \log\frac{p_\theta(\mathcal{M}_{t-1}^- \mid \mathcal{M}_t^-, \mathcal{P})}{p_{\mathrm{ref}}(\mathcal{M}_{t-1}^- \mid \mathcal{M}_t^-, \mathcal{P})} \right] \right) \right]. \tag{6}$$

The Diffusion-DPO loss is applied to align models with non-decomposable objectives using molecule-level preferences, which we will refer to as GLOBALDPO hereafter for clarity.

**LocalDPO** According to the decomposition in drug space, a molecule's probability factorizes over substructures (Guan et al., 2023b). As a result, we reformulate the Diffusion-DPO loss as:

$$\mathcal{L}_{\mathrm{DIFFUSION\text{-}DPO}} = -\mathbb{E}_{\substack{(\mathcal{P},\mathcal{M}^+,\mathcal{M}^-)\sim\mathcal{D},t\sim\mathcal{U}(0,T),\\ \mathcal{M}_t^+\sim q(\mathcal{M}_t^+|\mathcal{M}_0^+),\mathcal{M}_t^-\sim q(\mathcal{M}_t^-|\mathcal{M}_0^-)}}\left[ \log\sigma\left( \beta\sum_i^K\left[ \log\frac{p_\theta(\mathcal{M}_{t-1}^{(i)+} \mid \mathcal{M}_t^{(i)+}, \mathcal{P})}{p_{\mathrm{ref}}(\mathcal{M}_{t-1}^{(i)+} \mid \mathcal{M}_t^{(i)+}, \mathcal{P})} - \log\frac{p_\theta(\mathcal{M}_{t-1}^{(i)-} \mid \mathcal{M}_t^{(i)-}, \mathcal{P})}{p_{\mathrm{ref}}(\mathcal{M}_{t-1}^{(i)-} \mid \mathcal{M}_t^{(i)-}, \mathcal{P})} \right] \right) \right], \tag{7}$$

where $\mathcal{M}_t^{(i)+}$ and $\mathcal{M}_t^{(i)-}$ are the $i$-th decomposed substructure extracted from the winning and losing molecule, respectively. Here, the substructures of the preferred molecule are always considered the winning side, even though it is not always the case that they have better properties.

For decomposable objectives, we introduce decomposition into preference alignment by directly constructing preference pairs based on substructure-level properties, and define the LOCALDPO loss as:

$$\mathcal{L}_{\mathrm{LOCALDPO}} = -\mathbb{E}_{\substack{(\mathcal{P},\mathcal{M}^+,\mathcal{M}^-)\sim\mathcal{D},t\sim\mathcal{U}(0,T),\\ \mathcal{M}_t^+\sim q(\mathcal{M}_t^+|\mathcal{M}_0^+),\mathcal{M}_t^-\sim q(\mathcal{M}_t^-|\mathcal{M}_0^-)}}\left[ \log\sigma\big( \beta\sum_i^K \mathrm{sign}\big(r(\mathcal{M}^{(i)+}) - r(\mathcal{M}^{(i)-})\big)\big[ A^{(i)} \big] \big) \right],$$

$$\text{where } A^{(i)} = \log\frac{p_\theta(\mathcal{M}_{t-1}^{(i)+} \mid \mathcal{M}_t^{(i)+}, \mathcal{P})}{p_{\mathrm{ref}}(\mathcal{M}_{t-1}^{(i)+} \mid \mathcal{M}_t^{(i)+}, \mathcal{P})} - \log\frac{p_\theta(\mathcal{M}_{t-1}^{(i)-} \mid \mathcal{M}_t^{(i)-}, \mathcal{P})}{p_{\mathrm{ref}}(\mathcal{M}_{t-1}^{(i)-} \mid \mathcal{M}_t^{(i)-}, \mathcal{P})}, \tag{8}$$

and $r(\mathcal{M}^{(i)})$ represents the reward of the decomposed substructure $\mathcal{M}^{(i)}$. LOCALDPO can then be reformulated as GLOBALDPO on reconstructed molecule pairs assembled from substructures with higher- or lower-ranking properties. Please refer to Appendix A for the proof.

In multi-objective optimization, different objectives can interfere with each other, leading to suboptimal results. By allowing fine-grained guidance at the substructure level, LOCALDPO can alleviate conflicts in multi-objective optimization by offering more flexible and diverse optimization pathways, ultimately enhancing alignment performance with complex design goals.

Table 1: Summary of different properties of reference molecules and molecules generated by DECOMPDPO and other generative models (Gen.) and general purpose optimization methods (Opt.). (↑) / (↓) denotes a larger / smaller number is better. Top 2 results are highlighted with **bold text** and underlined text, respectively.

| | Methods | Vina Score (↓) | | Vina Min (↓) | | Vina Dock (↓) | | High Affinity (↑) | | QED (↑) | | SA (↑) | | Diversity (↑) | | Size | Success |
|---|---|---|---|---|---|---|---|---|---|---|---|---|---|---|---|---|---|
| | | Avg. | Med. | Avg. | Med. | Avg. | Med. | Avg. | Med. | Avg. | Med. | Avg. | Med. | Avg. | Med. | Avg. | Rate (↑) |
| | Reference | -6.36 | -6.46 | -6.71 | -6.49 | -7.45 | -7.26 | - | - | 0.48 | 0.47 | 0.73 | 0.74 | - | - | 22.8 | 25.0% |
| Gen. | LiGAN | - | - | - | - | -6.33 | -6.20 | 21.1% | 11.1% | 0.39 | 0.39 | 0.59 | 0.57 | 0.66 | 0.67 | 19.9 | 3.9% |
| | GraphBP | - | - | - | - | -4.80 | -4.70 | 14.2% | 6.7% | 0.43 | 0.45 | 0.49 | 0.48 | **0.79** | **0.78** | - | 0.1% |
| | AR | -5.75 | -5.64 | -6.18 | -5.88 | -6.75 | -6.62 | 37.9% | 31.0% | 0.51 | 0.50 | 0.63 | 0.63 | 0.70 | 0.70 | 17.7 | 7.1% |
| | Pocket2Mol | -5.14 | -4.70 | -6.42 | -5.82 | -7.15 | -6.79 | 48.4% | 51.0% | **0.56** | **0.57** | **0.74** | **0.75** | 0.69 | 0.71 | 17.7 | 24.4% |
| | TargetDiff | -5.47 | -6.30 | -6.64 | -6.83 | -7.80 | -7.91 | 58.1% | 59.1% | 0.48 | 0.48 | 0.58 | 0.58 | 0.72 | 0.71 | 24.2 | 10.5% |
| | IPDiff | -6.42 | -7.01 | -7.45 | -7.48 | -8.57 | -8.51 | 69.5% | 75.5% | 0.52 | 0.53 | 0.60 | 0.59 | 0.74 | 0.73 | 24.1 | 17.7% |
| | MolCRAFT | -6.59 | -7.04 | -7.27 | -7.26 | -7.92 | -8.01 | 59.0% | 62.6% | 0.50 | 0.51 | 0.69 | 0.68 | 0.72 | 0.73 | 26.8 | 22.7% |
| | DecompDiff* | -5.96 | -7.05 | -7.60 | -7.88 | -8.88 | -8.88 | 72.3% | 87.0% | 0.45 | 0.43 | 0.60 | 0.60 | 0.60 | 0.60 | 29.4 | 28.0% |
| Opt. | TAGMol | **-7.02** | **-7.77** | -7.95 | -8.07 | -8.59 | -8.69 | 69.8% | 76.4% | 0.55 | 0.56 | 0.56 | 0.56 | 0.69 | 0.70 | 24.7 | 11.1% |
| | DECOMPDPO | -6.13 | -7.54 | **-8.30** | **-8.57** | **-9.60** | **-9.68** | **85.8%** | **98.5%** | 0.48 | 0.46 | 0.67 | 0.67 | 0.63 | 0.62 | 31.6 | **43.9%** |

**DecompDpo** Based on the ideas introduced above, we define a unified loss that applies LOCALDPO to decomposable objectives and GLOBALDPO to non-decomposable objectives with a weighted sum:

$$\mathcal{L}_{\text{DecompDpo}} = \sum_{i \in \mathcal{Q}_{\text{Decomp}}} w_i \mathcal{L}_{\text{LocalDPO}}(i) + \sum_{j \in \mathcal{Q}_{\text{Non-Decomp}}} w_j \mathcal{L}_{\text{GlobalDPO}}(j), \tag{9}$$

where $\mathcal{Q}_{\text{Decomp}}$ and $\mathcal{Q}_{\text{Non-Decomp}}$ represent the set of decomposable and non-decomposable properties, and $w_i$, $w_j$ are weighting coefficients. This dual-granularity alignment accommodates both types of objectives and provides more precise control over the optimization process to meet the diverse requirements of molecule design.

### 3.3 Physically Constrained Optimization

An important aspect of preference alignment in drug design is ensuring the generated molecular conformations remain physically plausible. Inspired by Wu et al. (2022), we define physics-informed energy terms that penalize bonds and angles which deviate significantly from empirical values, formulated as:

$$\begin{aligned}
E_{\text{bond}} &= \sum_{i,j \in \mathcal{B}} \left( \max \left( 0, \left| L_{ij} - \mu^l_{v_i,v_j} \right| - 3\sigma^l_{v_i,v_j} \right) \right)^2, \\
E_{\text{angle}} &= \sum_{i,j,k \in \mathcal{A}} \left( \max \left( 0, \left| A_{ijk} - \mu^a_{v_i,v_j,v_k} \right| - 3\sigma^a_{v_i,v_j,v_k} \right) \right)^2,
\end{aligned} \tag{10}$$

where $\mathcal{B}$ denotes the set of bonds in the molecule, and $\mathcal{A}$ denotes the set of angles formed by two adjacent bonds in $\mathcal{B}$. Here, $L_{ij}$ is the bond length between atoms $i$ and $j$, and $\mu^l_{v_i,v_j}$, $\sigma^l_{v_i,v_j}$ are the expectation and standard deviation of bond lengths for those atom types, which are calculated from the training data. Similarly, for each angle $(i,j,k) \in \mathcal{A}$, $A_{ijk}$ measures the radian of the angle, and $\mu^a_{v_i,v_j,v_k}$, $\sigma^a_{v_i,v_j,v_k}$ represent the empirical angle statistics for the corresponding atom types. The overall energy term is defined as $r_{\text{constraint}} = E_{\text{bond}} + E_{\text{angle}}$. To constrain the model from learning unrealistic molecular conformations, we adjust the reward by penalizing it with this energy term: $r^*(\mathcal{M}, \mathcal{P}) = r(\mathcal{M}, \mathcal{P}) - \lambda r_{\text{constraint}}(\mathcal{M}, \mathcal{P})$, where $\lambda$ is a weighting factor that balances the original reward against structural validity. Additional evaluation in Appendix C demonstrates the effectiveness of this constraint in preserving reasonable molecular conformations during optimization.

### 3.4 Linear Beta Schedule

As shown in Equation (3), the parameter $\beta$ regulates how aggressively the model deviates from the reference distribution $p_{\text{ref}}$ in pursuit of higher rewards. During the diffusion process, earlier steps influence the subsequent ones. Larger $\beta$ in early diffusion steps can stabilize training and preserve adherence to the

Table 2: Summary of different properties of reference molecules and molecules generated by DECOMPDPO and other target-specific molecule optimization methods. (↑) / (↓) denotes a larger / smaller number is better. Top 2 results are highlighted with **bold text** and underlined text, respectively.

| Methods | Vina Score (↓) | | Vina Min (↓) | | Vina Dock (↓) | | High Affinity (↑) | | QED (↑) | | SA (↑) | | Diversity (↑) | | Success Rate (↑) |
|---------|------|------|------|------|------|------|------|------|------|------|------|------|------|------|------|
| | Avg. | Med. | Avg. | Med. | Avg. | Med. | Avg. | Med. | Avg. | Med. | Avg. | Med. | Avg. | Med. | |
| RGA | - | - | - | - | -8.01 | -8.17 | 64.4% | 89.3% | **0.57** | **0.57** | **0.71** | **0.73** | 0.41 | 0.41 | 46.2% |
| DecompOpt | -5.87 | -6.81 | -7.35 | -7.72 | -8.98 | -9.01 | 73.5% | 93.3% | 0.48 | 0.45 | 0.65 | 0.65 | 0.60 | 0.61 | **52.5%** |
| DECOMPDPO | **-7.27** | **-7.93** | **-8.91** | **-8.88** | **-9.90** | **-10.08** | **88.5%** | **100.0%** | 0.48 | 0.47 | 0.60 | 0.62 | **0.61** | **0.62** | 52.1% |

distribution learned from offline data with stronger regularization. As we progress through the final steps, where atom types and precise positions have a crucial impact on molecular properties, it becomes advantageous to relax this regularization to allow more aggressive optimization towards target objectives. To achieve this balance, we propose a linear beta schedule, $\beta_t = \frac{t}{T}\beta_T$, where $\beta_T$ is the maximum regularization weight at the initial stage $t = 0$ and decreases linearly as $t$ approaches $T$. This schedule effectively balances the influence of the reference model and optimization objectives throughout the diffusion process.

## 4 Experiments

We implement DECOMPDPO in two important pharmaceutical scenarios: (1) fine-tuning the model for molecule generation across diverse protein families, and (2) optimizing the model for a specific protein target.

### 4.1 Experimental Setup

**Dataset** We followed prior work (Luo et al., 2021; Peng et al., 2022; Guan et al., 2023a;b), using the Cross-Docked2020 dataset (Francoeur et al., 2020) to pre-train our reference model and evaluate the performance of DECOMPDPO. According to the protocol established by Luo et al. (2021), we filtered complexes to retain only those with high-quality docking poses (RMSD < 1Å) and diverse protein sequences (sequence identity < 30%), resulting in a refined dataset comprising 100,000 high-quality training complexes and 100 novel proteins for evaluation.

To fine-tune the model for general-purpose molecule generation, we first generate 10 candidate molecules for each training protein. Each molecule's favorability is measured by a multi-objective score $r_{multi} = \sum_{x_i \in X} x_i$, where $X$ is the set of normalized optimization objectives. For each protein, we select the top and bottom scoring molecules to form preference pairs, resulting in 63,092 valid pairs overall. To optimize the model for targeted molecule optimization, we sample 500 molecules for each test protein and construct preference pairs by similarly selecting the top and bottom 100 molecules according to their scores $r_{multi}$.

**Baselines** To assess the capability of DECOMPDPO in general-purpose molecule generation, we compare it with several representative generative models. **LiGAN** (Ragoza et al., 2022) employs a CNN-based variational autoencoder to encode receptor and ligand into a latent space, then generates ligands' atomic densities. Atom-based autoregressive models such as **AR** (Luo et al., 2021), **Pocket2Mol** (Peng et al., 2022), and **GraphBP** (Liu et al., 2022) update atom embeddings using a graph neural network (GNN). **TargetDiff** (Guan et al., 2023a) and **DecompDiff** (Guan et al., 2023b) are GNN-based diffusion models, with the latter incorporating decomposed priors. **IPDiff** (Huang et al., 2023) integrates protein-ligand interaction priors into both forward and reverse diffusion processes. **MolCRAFT** (Qu et al., 2024) introduces bayesian flow network to SBDD. We also compare with several general-purpose optimization methods. **TAGMol** (Dorna et al., 2024) optimizes continuous coordinates throughout training and sampling, while **AliDiff** (Gu et al., 2024) applies preference alignment, and **KGDiff** (Qian et al., 2024) uses an expert-network-based gradient guidance. Since AliDiff and KGDiff primarily focus on binding-affinity optimization, we compare with them under this setting.

To benchmark DECOMPDPO's capability in targeted optimization, we compare it with two strong baselines: **RGA** (Fu et al., 2022), a reinforced genetic algorithm that optimize molecules with a policy network through

evolutionary processes, and **DecompOpt** (Zhou et al., 2024a), which employs a conditional diffusion model to iteratively replace substructures for improved properties.

**Evaluation**    Following previous studies (Guan et al., 2023a; Luo et al., 2021; Ragoza et al., 2022), we evaluate molecules from two aspects: **target binding affinity and molecular properties**, and **molecular conformation**. We use AutoDock Vina to assess **target binding affinity**. *Vina Score* quantifies the direct binding affinity between a molecule and the target protein, *Vina Min* measures the affinity after local structural optimization via force fields, *Vina Dock* assesses the affinity after re-docking the ligand into the target protein, and *High Affinity* measures the proportion of generated molecules with higher *Vina Dock* score than that of reference ligands. For **molecular properties**, we calculate drug-likeness (*QED*) (Bickerton et al., 2012), synthetic accessibility (*SA*) (Ertl & Schuffenhauer, 2009), and *diversity*. Following Jin et al. (2020); Xie et al. (2021), the overall quality of generated molecules is evaluated by *Success Rate* (QED > 0.25, SA > 0.59, Vina Dock < -8.18). To evaluate **molecular conformation**, we compute Jensen-Shannon divergence (JSD) between the distributions of the generated molecules with reference ligands. We also evaluate median RMSD and energy difference of rigid fragments and the whole molecule before and after optimizing molecular conformations with Merck Molecular Force Field (MMFF) (Halgren, 1996). We provide more evaluation of the sanity of generated molecules in Appendix C.

**Implementation Details**    The bond-first noise schedule proposed by Peng et al. (2023) effectively addresses the inconsistency between atoms and bonds when using predicted bonds for molecule reconstruction. We adapt this noise schedule for DecompDiff, resulting in an enhanced model as our reference model, termed as DecompDiff*. We select *QED*, *SA*, and *Vina Minimize Score* as our optimization objectives. Please refer to Appendix B for more implementation details.

## 4.2 Main Results

**Molecule Generation**    We first evaluate the effectiveness of DECOMPDPO in optimizing models towards generating desirable molecules across various protein families. As shown in Table 1, DECOMPDPO significantly improves all metrics over the reference model, DecompDiff*, underscoring its effectiveness in multi-objective optimization. Notably, DECOMPDPO achieves the highest score for most binding-affinity-related metrics, confirming its robust ability to generate molecules that bind well across diverse protein families. Figure 3 shows examples of molecules generated by DECOMPDPO, illustrating that DECOMPDPO is capable of preserving realistic molecular conformations while achieving better scores. Additional visualization results are shown in Appendix C.

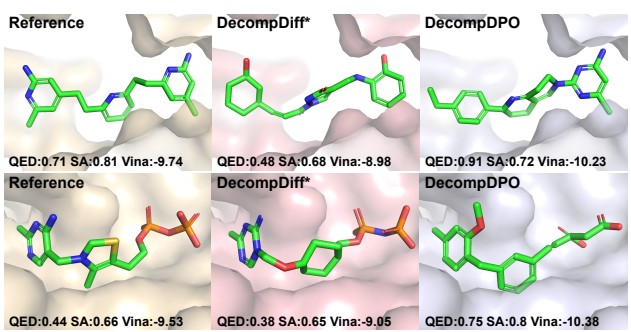

Figure 3: Visualization of reference binding ligands and the molecule generated by DECOMPDIFF* and DECOMPDPO on protein 4D7O (top) and 1UMD (bottom).

To demonstrate that DECOMPDPO preserves structurally realistic conformations while improving performance, we measure the Jensen–Shannon divergence (JSD) of all-atom pairwise distance distributions between generated molecules and reference ligands. As shown in Table 3, DECOMPDPO closely matches DecompDiff*, achieving the lowest JSD among all models evaluated. We further compute the median energy and RMSD differences pre- and post-MMFF optimization for both rigid fragments without rotatable bonds and whole molecules. The results indicate that DECOMPDPO generally performs on par with DecompDiff* and notably achieves the lowest whole-molecule median energy difference, underscoring DECOMPDPO 's capability in balancing property optimization with physically realistic structures. We provide additional distributions of these metrics, along with bond distance and bond angle JSD statistics, in Appendix C.

Table 4: Summary of results of single-objective optimization for affinity-related metrics. (↑) / (↓) denotes a larger / smaller number is better. The best result is highlighted with **bold text**.

| Method | Vina Score (↓) | | Vina Min (↓) | | Vina Dock (↓) | | High Affinity (↑) | | QED (↑) | | SA (↑) | | Diversity (↑) | | Size | Success |
|---|---|---|---|---|---|---|---|---|---|---|---|---|---|---|---|---|
| | Avg. | Med. | Avg. | Med. | Avg. | Med. | Avg. | Med. | Avg. | Med. | Avg. | Med. | Avg. | Med. | Avg. | Rate (↑) |
| KGDiff | **-8.04** | **-8.61** | -8.78 | -8.85 | -9.43 | -9.43 | 79.2% | 87.0% | **0.51** | **0.51** | 0.54 | 0.54 | **0.75** | **0.75** | 24.5 | 13.7% |
| AliDiff | -7.07 | -7.95 | -8.09 | -8.17 | -8.90 | -8.81 | 73.4% | 81.4% | 0.50 | 0.50 | 0.57 | 0.56 | 0.73 | 0.71 | 24.4 | 12.3% |
| DECOMPDPO | -6.52 | -8.04 | **-8.97** | **-9.15** | **-10.50** | **-10.29** | **91.8%** | **100.0%** | 0.46 | 0.43 | **0.67** | **0.67** | 0.70 | 0.70 | 31.7 | **46.8%** |

Table 5: Ablation study of decomposing DPO loss and linear beta schedule. (↑) / (↓) denotes a larger / smaller number is better. The best result is highlighted with **bold text**.

| Method | Vina Score (↓) | | Vina Min (↓) | | Vina Dock (↓) | | High Affinity (↑) | | QED (↑) | | SA (↑) | | Diversity (↑) | | Success |
|---|---|---|---|---|---|---|---|---|---|---|---|---|---|---|---|
| | Avg. | Med. | Avg. | Med. | Avg. | Med. | Avg. | Med. | Avg. | Med. | Avg. | Med. | Avg. | Med. | Rate (↑) |
| w/ Constant Beta Weight | -5.97 | -7.14 | -7.78 | -8.04 | -9.04 | -9.09 | 74.9% | 91.8% | 0.46 | 0.44 | 0.62 | 0.62 | 0.61 | 0.61 | 32.1% |
| Molecule-level DPO | -5.84 | -7.10 | -7.75 | -8.03 | -9.06 | -9.10 | 76.2% | 92.3% | **0.49** | **0.47** | 0.63 | 0.64 | 0.62 | 0.62 | 35.5% |
| DECOMPDPO | **-6.13** | **-7.54** | **-8.30** | **-8.57** | **-9.60** | **-9.68** | **85.8%** | **98.5%** | 0.48 | 0.46 | **0.67** | **0.67** | **0.63** | 0.62 | **43.9%** |

**Targeted Optimization** To evaluate DECOMPDPO 's effectiveness in target-specific molecule optimization, we further optimize the fine-tuned model for each target protein in the test set. As shown in Table 2, DECOMPDPO outperforms all baselines in affinity-related metrics and achieves a *Success Rate* comparable to DecompOpt, illustrating its strong ability to optimize ligands for specific protein targets. In practice, DECOMPDPO can also be integrated into DecompOpt, potentially unlocking even greater improvements by combining preference alignment with iterative optimization.

Table 3: Summary of conformation related metrics of molecules (Mol.) and corresponding rigid fragments (RF.) generated by DECOMPDPO and other diffusion models. The top 2 results are highlighted with **bold text** and underlined text.

| | TargetDiff | IPDiff | DecompDiff* | DecompDPO |
|---|---|---|---|---|
| JSD - All Atom | 0.09 | 0.08 | **0.07** | **0.07** |
| Energy Diff - RF. | 1355.94 | 1459.45 | 39.39 | **38.38** |
| Energy Diff - Mol. | 6116.37 | 21431.71 | 8833.80 | **672.02** |
| RMSD - RF. | 0.13 | 0.14 | 0.13 | **0.11** |
| RMSD - Mol. | **1.02** | 1.04 | 1.10 | 1.08 |

### 4.3 Ablation Studies

**Single-Objective Optimization** To further demonstrate DECOMPDPO 's effectiveness in binding-affinity optimization, we use LOCALDPO for *Vina Minimize* optimization. As shown in Table 4, DECOMPDPO achieves the highest scores on most affinity-related metrics among general-purpose binding-affinity optimization methods, with improvements of 9.4% in *Vina Score*, 18.0% in *Vina Minimize*, and 18.2% in *Vina Dock* over the reference model. Notably, DECOMPDPO also achieves improvements in *QED* and *SA*, which we attribute to the negative correlations (-0.12 and -0.11) between substructure-level *Vina Minimize* differences and corresponding molecule-level *QED* and *SA* differences in our training preference pairs. Additionally, DECOMPDPO achieves 78.9% *Complete Rate*, defined as the percentage of valid and connected molecules among all generated molecules, indicating that its performance boost does not come at the cost of general molecular properties.

**Benefits of Decomposed Preference** Our primary hypothesis is that decomposing the optimization objectives improves training efficiency by offering more flexibility in preference selection, particularly in multi-objective settings. We verify this claim by fine-tuning the reference model with only molecule-level preference pairs for all optimization objectives, which we term as Molecule-level DPO. As shown in Table 5, DECOMPDPO outperforms Molecule-level DPO across all metrics, validating that decomposed preference could provide greater flexibility and diversity in preference selection for enhanced optimization effectiveness.

**Benefits of Linear Beta Schedule** We also examine the effectiveness of the linear beta schedule by comparing it with a constant-$\beta$ baseline for molecule generation. As shown in Table 5, the linear schedule consistently enhances all reported metrics, illustrating its effectiveness in balancing adherence to the reference

distribution with high-reward optimization, leading to more efficient preference alignment for diffusion models.

## 5 Conclusion

In this work, we introduced preference alignment to SBDD, developing DECOMPDPO to align pre-trained diffusion models with multi-granularity preference, which provides more flexibility during the optimization process. The physics-informed energy term penalizing the reward is beneficial for maintaining reasonable molecular conformations during optimization. The linear beta schedule effectively improves optimization efficiency by progressively reducing regularization during the diffusion process. DECOMPDPO shows promising results in molecule generation and molecule optimization, highlighting its ability to meet practical needs of the pharmaceutical industry.

## Broader Impact Statement

Our contributions to structure-based drug design have the potential to significantly accelerate the drug discovery process, thereby transforming the pharmaceutical research landscape. Furthermore, the versatility of our approach allows for its application in other domains of computer-aided design, including, but not limited to, protein design, material design, and chip design. While the potential impacts are ample, we underscore the importance of implementing our methods responsibly to prevent misuse and potential harm. Hence, diligent oversight and ethical considerations remain paramount in ensuring the beneficial utilization of our techniques.

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

## A  Theoretical Analysis

According to DecompDiff (Guan et al., 2023b), the reverse diffusion probability of a molecule can be factorized over its substructures as:

$$
\begin{aligned}
p(\mathcal{M}_{t-1} \mid \mathcal{M}_t, \mathcal{M}_0) &= p(\mathbf{x}_{t-1} \mid \mathbf{x}_t, \mathbf{x}_0) \cdot p(\mathbf{v}_{t-1} \mid \mathbf{v}_t, \mathbf{v}_0) \cdot p(\mathbf{b}_{t-1} \mid \mathbf{b}_t, \mathbf{b}_0) \\
&= \prod_{k=1}^{K} \left( \prod_{i=1}^{N_M} \mathbf{H}_{ik} \cdot p(\mathbf{x}_{t-1}^i \mid \mathbf{x}_t^i, \mathbf{x}_0^i) \cdot p(\mathbf{v}_{t-1}^i \mid \mathbf{v}_t^i, \mathbf{v}_0^i) \right) \prod_{i=1}^{N_M} \prod_{j=1}^{N_M} \mathbf{H}_{ik} \cdot \mathbf{H}_{jk} \cdot p(\mathbf{b}_{t-1}^{ij} \mid \mathbf{b}_t^{ij}, \mathbf{b}_0^{ij}) \\
&= \prod_{k=1}^{K} p(\mathcal{M}_{t-1}^{(k)} \mid \mathcal{M}_t^{(k)}, \mathcal{M}_0^{(k)}).
\end{aligned}
\tag{11}
$$

Under the assumption of decomposable molecular probability, we can then reformulate LOCALDPO loss as:

$$
\begin{aligned}
\mathcal{L}_{\text{LOCALDPO}} &= -\mathbb{E}_{\text{data}}\Big[\log\sigma\big(\beta\sum_{i}^{K}\text{sign}(r(\mathcal{M}^{(i)+})-r(\mathcal{M}^{(i)-}))\big[\log\frac{p_\theta(\mathcal{M}_{t-1}^{(i)+}\mid\mathcal{M}_t^{(i)+},\mathcal{P})}{p_{\text{ref}}(\mathcal{M}_{t-1}^{(i)+}\mid\mathcal{M}_t^{(i)+},\mathcal{P})} - \log\frac{p_\theta(\mathcal{M}_{t-1}^{(i)-}\mid\mathcal{M}_t^{(i)-},\mathcal{P})}{p_{\text{ref}}(\mathcal{M}_{t-1}^{(i)-}\mid\mathcal{M}_t^{(i)-},\mathcal{P})}\big]\big)\Big] \\
&= -\mathbb{E}_{\text{data}}\Big[\log\sigma\big(\beta\sum_{i}^{K}\big[\log\frac{p_\theta(\mathcal{M}_{t-1}^{(i')+}\mid\mathcal{M}_t^{(i')+},\mathcal{P})}{p_{\text{ref}}(\mathcal{M}_{t-1}^{(i')+}\mid\mathcal{M}_t^{(i')+},\mathcal{P})} - \log\frac{p_\theta(\mathcal{M}_{t-1}^{(i')-}\mid\mathcal{M}_t^{(i')-},\mathcal{P})}{p_{\text{ref}}(\mathcal{M}_{t-1}^{(i')-}\mid\mathcal{M}_t^{(i')-},\mathcal{P})}\big]\big)\Big] \\
&= -\mathbb{E}_{\text{data}}\Big[\log\sigma\big(\beta\big[\log\frac{p_\theta(\mathcal{M}_{t-1}'^{+}\mid\mathcal{M}_t'^{+},\mathcal{P})}{p_{\text{ref}}(\mathcal{M}_{t-1}'^{+}\mid\mathcal{M}_t'^{+},\mathcal{P})} - \log\frac{p_\theta(\mathcal{M}_{t-1}'^{-}\mid\mathcal{M}_t'^{-},\mathcal{P})}{p_{\text{ref}}(\mathcal{M}_{t-1}'^{-}\mid\mathcal{M}_t'^{-},\mathcal{P})}\big]\big)\Big],
\end{aligned}
\tag{12}
$$

where $\mathbb{E}_{\text{data}} = \mathbb{E}_{\substack{(\mathcal{P},\mathcal{M}^+,\mathcal{M}^-)\sim\mathcal{D},t\sim\mathcal{U}(0,T),\\ \mathcal{M}_t^+\sim q(\mathcal{M}_t^+\mid\mathcal{M}_0^+),\mathcal{M}_t^-\sim q(\mathcal{M}_t^-\mid\mathcal{M}_0^-)}}$, $\mathcal{M}^{(i)+}$ and $\mathcal{M}^{(i)-}$ denote the $i$-th substructures decomposed from the original winning and losing molecules, $\mathcal{M}^{(i')+}$ and $\mathcal{M}^{(i')-}$ refer to the $i$-th substructures with higher and lower individual property, $\mathcal{M}'^{+}$ and $\mathcal{M}'^{-}$ denote the molecules reconstructed from substructures with higher and lower property scores across all decomposed substructures. We can then reinterpret LOCALDPO as GLOBALDPO over recombined molecule pairs with higher and lower properties than original molecules, enabling the model to learn optimization over more extreme cases.

## B Implementation Details

### B.1 Featurization

Following DecompDiff (Guan et al., 2023b), we characterize each protein atom using a set of features: a one-hot indicator of the element type (H, C, N, O, S, Se), a one-hot indicator of the amino acid type to which the atom belongs, a one-dimensional indicator denoting whether the atom belongs to the backbone, and a one-hot indicator specifying the arm/scaffold region. We define the part of proteins that lies within 10Å of any atom of the ligand as the pocket. Similarly, a protein atom is assigned to the arm region if it lies within a 10Å radius of any arm; otherwise, it is categorized under the scaffold region. The ligand atom is characterized with a one-hot indicator of element type (C, N, O, F, P, S, Cl) and a one-hot arm/scaffold indicator. The partition of arms and scaffold is predefined by a decomposition algorithm proposed by DecompDiff.

We use two types of message-passing graphs to model the protein-ligand complex: a $k$-nearest neighbors (knn) graph for all atoms (we choose $k = 32$ in all experiments) and a fully-connected graph for ligand atoms only. In the knn graph, edge features are obtained from the outer product of the distance embedding and the edge type. The distance embedding is calculated using radial basis functions centered at 20 points between 0Å and 10Å. Edge types are represented by a 4-dimensional one-hot vector, categorizing edges as between ligand atoms, protein atoms, ligand-protein atoms or protein-ligand atoms. For the fully-connected ligand graph, edge features include a one-hot bond type indicator (non-bond, single, double, triple, aromatic) and a feature indicating whether the bonded atoms belong to the same arm or scaffold.

### B.2 Model Details

Our base model used in DECOMPDPO is the model proposed by Guan et al. (2023b), incorporating the bond first noise schedule presented by Peng et al. (2023). Specifically, the noise schedule is defined as follows:

$$
\begin{aligned}
s &= \frac{s_T - s_1}{\text{sigmoid}(-w) - \text{sigmoid}(w)} \\
b &= \frac{s_1 + s_T + s}{2} \\
\bar{\alpha}_t &= s \cdot \text{sigmoid}(-w(2t/T - 1)) + b
\end{aligned}
$$

For atom types, the parameters of noise schedule are set as $s_1 = 0.9999$, $s_T = 0.0001$, $w = 3$. For bond types, a two-stage noise schedule is employed: in the initial stage ($t \in [1, 600]$), bonds are rapidly diffused with parameters $s_1 = 0.9999$, $s_T = 0.001$, $w = 3$. In the subsequent stage ($t \in [600, 1000]$), the parameters are set as $s_1 = 0.001$, $s_T = 0.0001$, $w = 2$. The schedules of atom and bond type are shown in Figure 4.

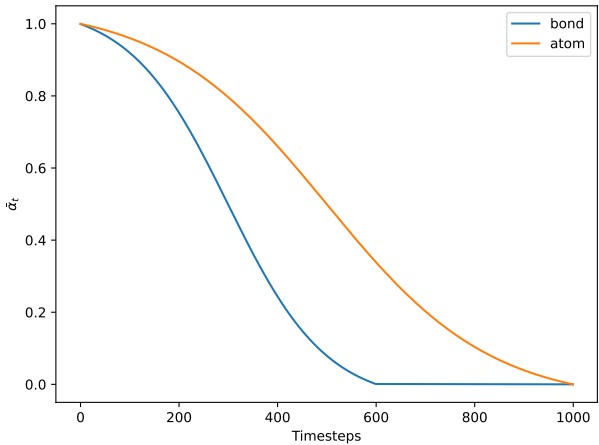

Figure 4: Noise schedule of atom and bond types.

### B.3 Molecular Fragmentation

Following DecompDiff (Guan et al., 2023b), we fragment a molecule into arms and scaffold using RDKit and Alphaspace2 (Katigbak et al., 2020) toolkit. Specifically, subpockets for the target protein are extracted using Alphaspace2 and ligands are decomposed into fragments using BRICS. Then terminal fragments with only one connection site are assigned to subpockets by a linear sum assignment. Arms centers are defined as the centroids of terminal fragments and any remaining subpockets, and scaffold center is defined as the farthest fragment from all arm centers. Finally, the nearest neighbor clustering is performed to tag fragments as arms or the scaffold.

### B.4 Training Details

**Pre-training**   We use Adam (Kingma & Ba, 2014) for pre-training, with `init_learning_rate=0.0004` and `betas=(0.95,0.999)`. The learning rate is scheduled to decay exponentially with a factor of 0.6 with `minimize_learning_rate=1e-6`. The learning rate is decayed if there is no improvement for the validation loss in 10 consecutive evaluations. We set `batch_size=8` and `clip_gradient_norm=8`. During training, a small Gaussian noise with a standard deviation of 0.1 to protein atom positions is added as data augmentation. To balance the magnitude of different losses, the reconstruction losses of atom and bond type are multiplied by weights $\gamma_v = 100$ and $\gamma_b = 100$, respectively. We perform evaluations every 2000 training steps. The model is pre-trained on a single NVIDIA A6000 GPU, and it could converge within 21 hours and 170k steps.

**Fine-tuning and Optimizing**   For both fine-tuning and optimizing model with DECOMPDPO, we use the Adam optimizer with `init_learning_rate=1e-6` and `betas=(0.95,0.999)`. We maintain a constant learning rate throughout both processes. We set `batch_size=4` and `clip_gradient_norm=8`. Consistent with pre-training, Gaussian noise is added to protein atom positions, and we use a weighted reconstruction loss. For fine-tuning the model for molecule generation, we set $\beta_T = 0.001$ and trained for 30,000 steps on one NVIDIA A40 GPU. For molecular optimization, we set $\beta_T = 0.02$ and trained for 20,000 steps on one NVIDIA V100 GPU. We perform evaluation every 1,000 steps.

### B.5 Experiment Details

The scoring function for selecting training molecules is defined as $S = QED + SA + Vina\_Min/(-12)$. *Vina Minimize Score* is divided by -12 to ensure that it generally ranges between 0 and 1. For molecule generation, we exclude molecules that cannot be decomposed or reconstructed, resulting in a total of 63,092 preference pairs available for fine-tuning. The weight of each objective in multi-objective optimization is set to 1. In molecular optimization, to ensure that the model maintains a desirable completion rate, we include an additional 50 molecules that failed in reconstruction in the losing side of preference pairs. To tailor the optimization to a specific protein, the weights of the optimization objectives are defined as $w_x = e^{-(x-x_s)}$,

where $x$ is the mean property of the generated molecules and $x_s$ is the threshold of the property used in *Success Rate*. For both molecule generation and molecular optimization, we employ the same *Opt Prior* used in DecompDiff. *Opt Prior* is defined as a mixture of *Ref Prior*, which is determined by the reference ligand, and *Pocket Prior*, which is defined by a prior generation algorithm using AlphaSpace2 (Katigbak et al., 2020), depending on whether Ref Prior passes the *Success threshold*. The $\lambda$ used for penalizing rewards with energy terms proposed in Section 3.3 is set to 0.1.

In evaluating the performance of DECOMPDPO, for each checkpoint, we generate 100 molecules for the molecule generation task and 20 molecules for the molecular optimization task across each target protein in the test set. For both molecule generation and optimization, we select the checkpoint with the highest *weighted Success Rate*, which is defined as the product of the *Success Rate* and the *Complete Rate*.

## C  Additional Results

### C.1  Full Evaluation Results

Table 6: Jensen-Shannon Divergence of the bond distance distribution between the generated molecules and the reference molecule by bond type, with a lower value indicating better. "-", "=", and ":" represent single, double, and aromatic bonds, respectively. The top 2 results are highlighted with **bold text** and underlined text.

| Bond | liGAN | GraphBP | AR | Pocket2Mol | TargetDiff | DecompDiff | IPDiff | DECOMPDPO |
|------|-------|---------|-----|-----------|-----------|-----------|--------|-----------|
| C−C | 0.601 | 0.368 | 0.609 | 0.496 | 0.369 | **0.359** | 0.451 | 0.535 |
| C=C | 0.665 | 0.530 | 0.620 | 0.561 | **0.505** | 0.537 | 0.530 | 0.546 |
| C−N | 0.634 | 0.456 | 0.474 | 0.416 | 0.363 | **0.344** | 0.411 | 0.404 |
| C=N | 0.749 | 0.693 | 0.635 | 0.629 | **0.550** | 0.584 | 0.567 | 0.598 |
| C−O | 0.656 | 0.467 | 0.492 | 0.454 | 0.421 | **0.376** | 0.489 | 0.411 |
| C=O | 0.661 | 0.471 | 0.558 | 0.516 | 0.461 | 0.374 | 0.431 | **0.319** |
| C:C | 0.497 | 0.407 | 0.451 | 0.416 | 0.263 | 0.251 | **0.221** | 0.281 |
| C:N | 0.638 | 0.689 | 0.552 | 0.487 | **0.235** | 0.269 | 0.255 | 0.265 |

Table 7: Jensen-Shannon Divergence of the bond angle distribution between the generated molecules and the reference molecule by angle type, with a lower value indicating better. The top 2 results are highlighted with **bold text** and underlined text.

| Bond | liGAN | GraphBP | AR | Pocket2Mol | TargetDiff | DecompDiff | IPDiff | DECOMPDPO |
|------|-------|---------|-----|-----------|-----------|-----------|--------|-----------|
| CCC | 0.598 | 0.424 | 0.340 | 0.323 | 0.328 | **0.314** | 0.402 | 0.488 |
| CCO | 0.637 | 0.354 | 0.442 | 0.401 | 0.385 | **0.324** | 0.451 | 0.428 |
| CNC | 0.604 | 0.469 | 0.419 | **0.237** | 0.367 | 0.297 | 0.407 | 0.360 |
| OPO | 0.512 | 0.684 | 0.367 | 0.274 | 0.303 | 0.217 | 0.388 | **0.185** |
| NCC | 0.621 | 0.372 | 0.392 | 0.351 | 0.354 | **0.294** | 0.399 | 0.375 |
| CC=O | 0.636 | 0.377 | 0.476 | 0.353 | 0.356 | **0.259** | 0.363 | 0.265 |
| COC | 0.606 | 0.482 | 0.459 | **0.317** | 0.389 | 0.339 | 0.463 | 0.427 |

**Molecular Conformation**   To provide a more comprehensive evaluation of molecular conformations, we compute the JSD of distances for different types of bonds and angles between molecules from generative models and reference molecules. As shown in Table 6 and Table 7, DECOMPDPO generally achieves comparable JSD values to those of DecompDiff, demonstrating that DECOMPDPO generally maintains desirable molecular conformations during preference alignment.

In Figure 5, we compare the pairwise distance distributions between generated molecules and reference ligands, along with their corresponding Jensen–Shannon divergence (JSD). DECOMPDPO and DecompDiff* both achieve low JSD scores, indicating minimal deviation from the distribution of real molecules. We further investigate the structural quality of generated molecules by examining both rigid fragments and whole molecules before and after MMFF optimization. As shown in Figure 6, DECOMPDPO shows lower or comparable median energy differences relative to DecompDiff*, and consistently lower RMSD differences across various fragment sizes, validating its improvements in optimization objectives coincide with maintaining or

improving conformational quality. In Figure 7, the median RMSD and energy differences for whole molecules further confirm DECOMPDPO preserves physically realistic geometries.

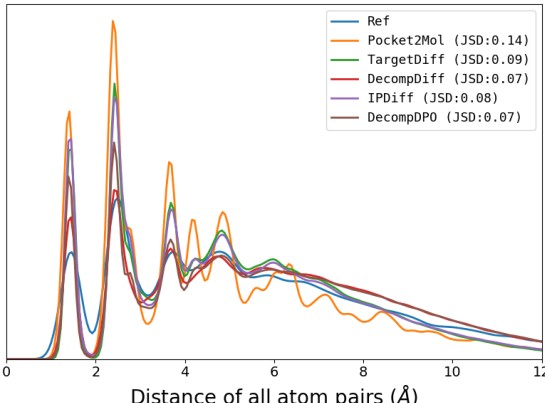

Figure 5: Compare pairwise distance distributions between all atoms in generated molecules and reference molecules from the test set. Jensen-Shannon divergence (JSD) between two distributions is reported.

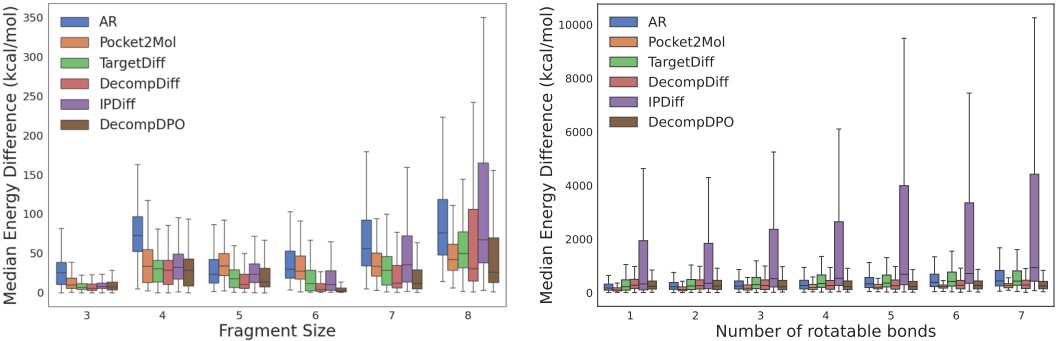

Figure 6: Median energy difference for rigid fragments (left) and generated molecules (right) before and after optimizing with the Merck Molecular Force Field.

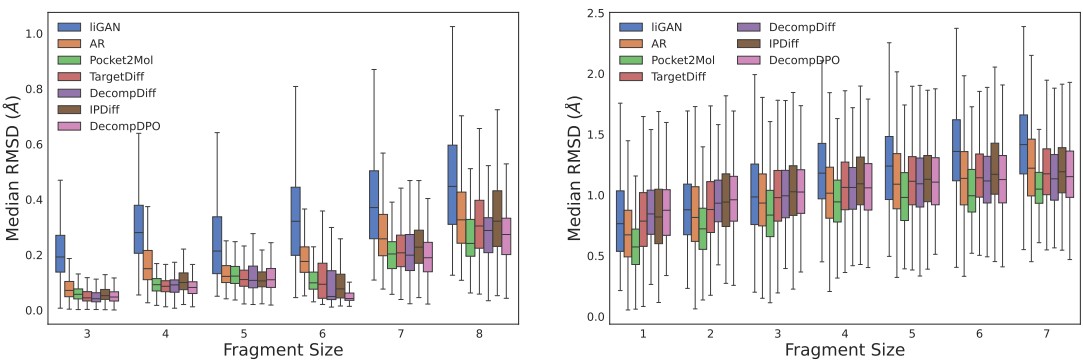

Figure 7: Median RMSD for rigid fragments (left) and generated molecules (right) before and after optimizing with the Merck Molecular Force Field.

**Molecular Properties** To provide a comprehensive evaluation, we have expanded our evaluation metrics beyond those discussed in Section 4.1, which primarily focus on molecular properties and binding affinities.

To assess the model's efficacy in designing novel and valid molecules, we calculate the following additional metrics:

- *Complete Rate* is the percentage of generated molecules that are connected and valid, which is defined by RDKit.

- *Novelty* is defined as the ratio of generated molecules that are different from the reference ligand of the corresponding pocket in the test set.

- *Similarity* is the Tanimoto Similarity between generated molecules and the corresponding reference ligand.

- *Uniqueness* is the proportion of unique molecules among generated molecules.

Table 8: Summary of the models' ability in designing novel and valid molecules. (↑) / (↓) denotes a larger / smaller number is better.

| | Methods | Complete Rate (↑) | Novelty (↑) | Similarity (↓) | Uniqueness (↑) |
|---|---|---|---|---|---|
| Generate | LiGAN | 99.11% | 100% | 0.22 | 87.82% |
| | AR | 92.95% | 100% | 0.24 | 100% |
| | Pocket2Mol | 98.31% | 100% | 0.26 | 100% |
| | TargetDiff | 90.36% | 100% | 0.30 | 99.63% |
| | DECOMPDIFF* | 72.82% | 100% | 0.27 | 99.58% |
| | DECOMPDPO | 74.05% | 100% | 0.26 | 99.57% |
| Optimize | RGA | - | 100% | 0.37 | 96.82% |
| | DecompOpt | 71.55% | 100% | 0.36 | 100% |
| | DECOMPDPO | 65.05% | 100% | 0.26 | 99.63% |

As reported in Table 8, in molecule generation, DECOMPDPO fine-tuned model achieves better *Complete Rate* and *Similarity* compared to the base model. In molecular optimization, DECOMPDPO maintains a relatively acceptable *Complete Rate* and the lowest similarity among all optimization methods.

Recent works (Qu et al., 2024; Schneuing et al.) have further emphasized the importance of ensuring the physical realism beyond optimizing for molecular properties. To assess the performance of DECOMPDPO in generating valid molecules, we use PoseBuster (Buttenschoen et al., 2024) and PoseCheck (Harris et al., 2023) to compute several additional metrics:

- *Strain Energy* reflects the internal energy cost of the generated conformation relative to a relaxed state. We report the 25%, 50%, and 75% quartiles to give a more global view of ligand stability.

- *Clash* is the average number of steric clashes between ligand atoms and protein residues, thus highlighting potential conflicts in the binding pose.

- *HB Donors* and *HB Acceptors* quantify the number of hydrogen-bond interactions formed by the ligand, while *Hydrophobic* and *vdWs* are the number of non-polar and van der Waals interactions, which often correlate with effective binding.

- *PB-Valid* reports the percentage of generated molecules that pass POSEBUSTER's built-in validation checks, accounting for geometry and potential clashes.

As shown in Table 9, DECOMPDPO exhibits the most favorable Strain Energy distributions among all the diffusion-based generative models, indicating that its generated ligands tend to adopt more stable conformations. Although the average number of steric clashes (*Clash*) remains somewhat high, DECOMPDPO slightly improves upon DECOMPDIFF, suggesting that the remaining anomalies can be mitigated in future work by refining reference distributions. In addition, DECOMPDPO shows an increase in the PB valid rate, confirming that its improvements in molecular design do not come at the cost of producing invalid structures.

Table 9: Summary of different properties of reference molecules and molecules generated by DECOMPDPO and other generative models. (↑) / (↓) denotes a larger / smaller number is better. Top 2 results are highlighted with **bold text** and underlined text, respectively.

| | Strain Energy (↓) | | | Clash (↓) | HB Donors (↑) | HB Acceptor (↑) | Hydrophobic (↑) | vdWs (↑) | PB-Valid (↑) |
|---|---|---|---|---|---|---|---|---|---|
| | 25% | 50% | 75% | Avg. | Avg. | Avg. | Avg. | Avg. | (%) |
| Reference | 34 | 107 | 196 | 5.51 | 0.87 | 1.42 | 5.06 | 6.61 | 95.0% |
| AR | 259 | 595 | 2286 | **4.49** | 0.51 | 0.90 | 3.78 | 5.54 | 55.6% |
| Pocket2Mol | **102** | **189** | **374** | 6.24 | 0.32 | 0.63 | 4.53 | 5.25 | **73.1%** |
| TargetDiff | 369 | 1243 | 13871 | 10.84 | **0.63** | 0.98 | 5.43 | 7.92 | 50.8% |
| DecompDiff | 162 | 354 | 802 | 15.42 | 0.59 | **1.48** | 7.96 | **11.06** | 54.6% |
| DecompDPO | 141 | 323 | 724 | 15.05 | 0.43 | 1.31 | **8.25** | 11.01 | 56.6% |

Overall, these results reinforce that preference alignment in DECOMPDPO not only improves optimization objectives but also helps maintain chemically and structurally reasonable molecular conformations.

We also draw boxplots to provide confidence intervals for the performance in molecule generation, which are shown in Figure 8.

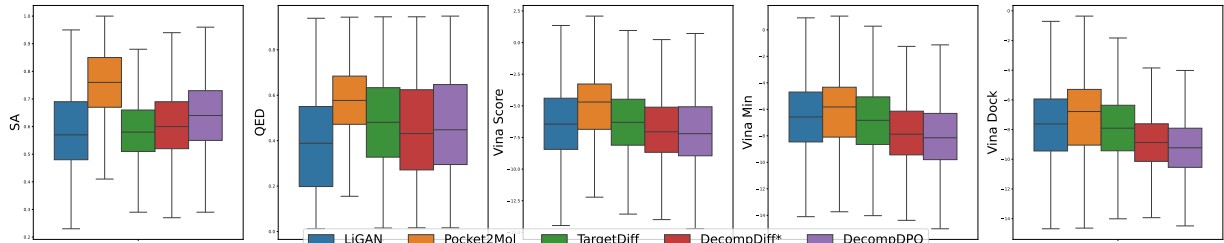

Figure 8: Boxplots of QED, SA, Vina Score, Vina Minimize, and Vina Dock of molecules generated by DECOMPDPO and other generative models.

To further illustrate the potency of the generated molecules, we draw a scatter plot of heavy atom numbers versus Vina Dock score to demonstrate the effect of heavy atom numbers on the binding affinity of generated molecules.

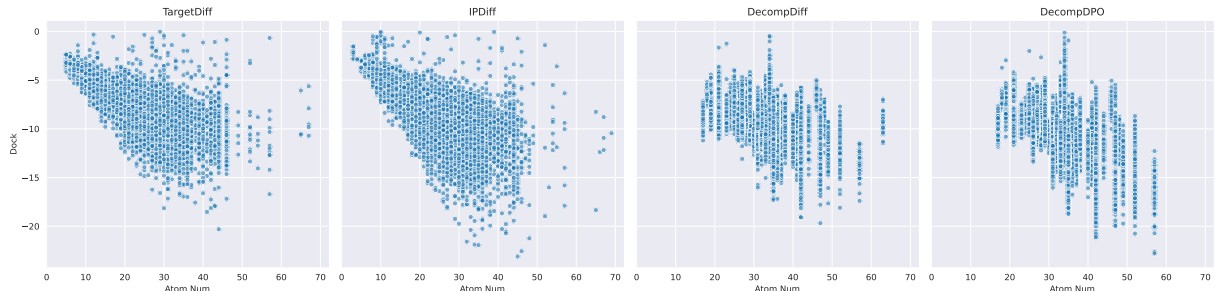

Figure 9: Scatter Plots of heavy atom numbers versus Vina Dock scores for TargetDiff, IPDiff, DecompDiff, and DECOMPDPO.

## C.2 Benefits of Physics-informed Energy Terms

As shown in Table 10, DecompDPO achieves comparable optimization results with and without using physics-informed energy terms. However, as suggested in Table 11, without using physics-informed energy terms, the energy difference of both whole molecule and rigid fragments significantly increases, demonstrating the necessity of physics-informed energy terms in maintaining reasonable conformations during optimization.

Table 10: Summary of results of DECOMPDPO with and without using physics-informed energy terms. (↑) / (↓) denotes a larger / smaller number is better. The better result is highlighted with bold text.

| Method | Vina Score (↓) | | Vina Min (↓) | | Vina Dock (↓) | | High Affinity (↑) | | QED (↑) | | SA (↑) | | Diversity (↑) | | Success |
|---|---|---|---|---|---|---|---|---|---|---|---|---|---|---|---|
| | Avg. | Med. | Avg. | Med. | Avg. | Med. | Avg. | Med. | Avg. | Med. | Avg. | Med. | Avg. | Med. | Rate (↑) |
| w/o phys | **-6.17** | -7.37 | -8.07 | -8.32 | -9.15 | -9.36 | 80.7% | 95.7% | 0.47 | 0.44 | 0.65 | 0.65 | 0.61 | 0.62 | 38.9% |
| DECOMPDPO | -6.13 | **-7.54** | **-8.30** | **-8.57** | **-9.60** | **-9.68** | **85.8%** | **98.5%** | **0.48** | **0.46** | **0.67** | **0.67** | **0.63** | 0.62 | **43.9%** |

Table 11: Evaluation of physical realism with and without physics-informed energy terms. Lower energy differences and RMSD indicate better structural consistency.

| Method | JSD Dist (↓) | Energy Diff - rigid frags (↓) | Energy Diff - Mol (↓) | RMSD - rigid frags (↓) | RMSD - Mol (↓) |
|---|---|---|---|---|---|
| w/o phys | 0.07 | 48.00 | 887.15 | 0.12 | 1.10 |
| DecompDPO | 0.07 | **38.38** | **672.02** | **0.11** | **1.08** |

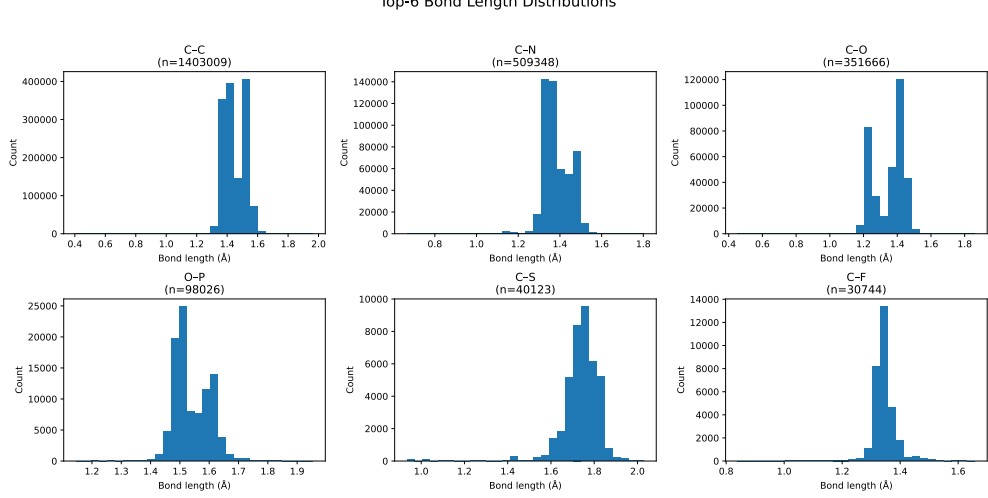

Figure 10: Histogram of bond lengths for the six most frequent atom pairs.

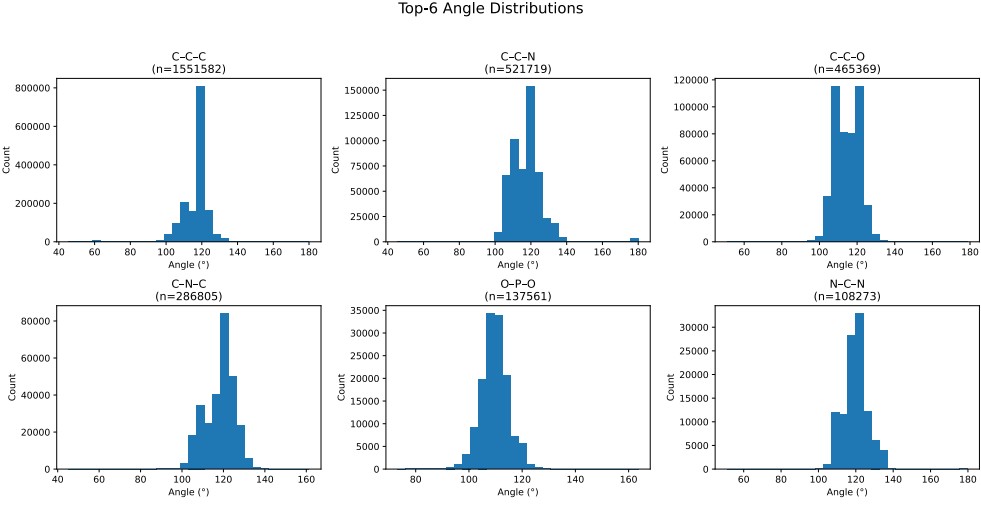

Figure 11: Histogram of bond angles for the six most frequent atom triplets.

To further substantiate the rationale for our physics-informed energy constraint, we visualize the training-set distributions of the six most frequent bond-length substructures and bond-angle substructures in Figure 10 and Figure 11, which we adopt as empirical priors to penalize geometries that deviate from physically realistic ranges during optimization.

### C.3  Evidence for Decomposability of Properties

As illustrated in Section 3.2, *QED* and *SA* are non-decomposable due to the non-linear processes involved in their calculations. We validate this non-decomposability on our training set. As shown in Figure 12, the Pearson correlation coefficients between the properties of molecules and the sum of the properties of their decomposed substructures are very low, not exceeding 0.1. These results indicate that substructures with higher *QED* or *SA* do not necessarily lead the molecule to have better properties. Therefore, we choose molecule-level preferences for *QED* and *SA*.

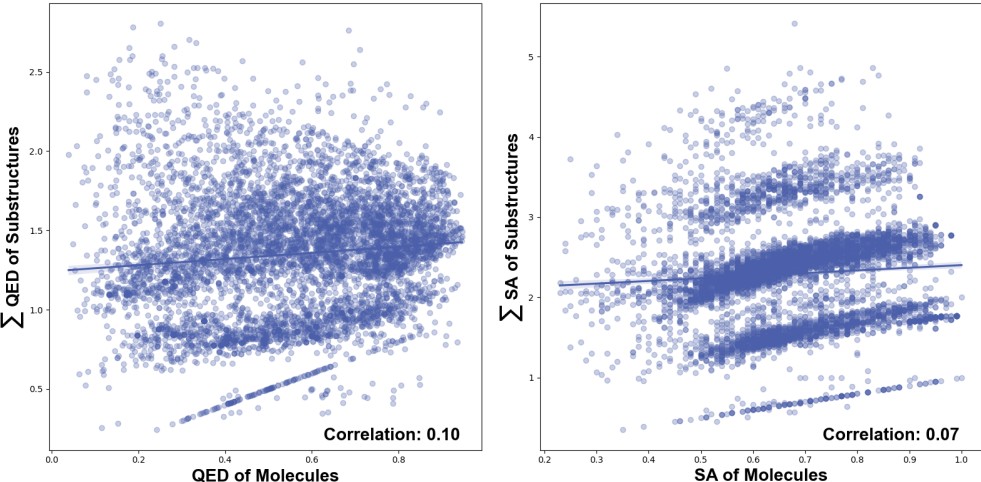

Figure 12: The Pearson correlation between molecule's and sum of substructures' SA (left) / QED (right) Scores in the training dataset.

### C.4  Trade-off in Multi-objective Optimization

Given the multiple objectives in DECOMPDPO, inherent trade-offs between different properties are unavoidable. In molecular optimization, as illustrated in Figure 13, molecules generated by DECOMPDPO exhibit significantly improved properties compared to those generated by DecompDiff. However, DECOMPDPO encounters a notable trade-off between optimizing the *Vina Minimize Score* and *SA*.

### C.5  Training Set Distribution

We further provided winning and losing molecules' distribution, as well as the distribution of the difference in QED, SA, and Vina Minimize in the training set, as shown in Figure 14 and Figure 15.

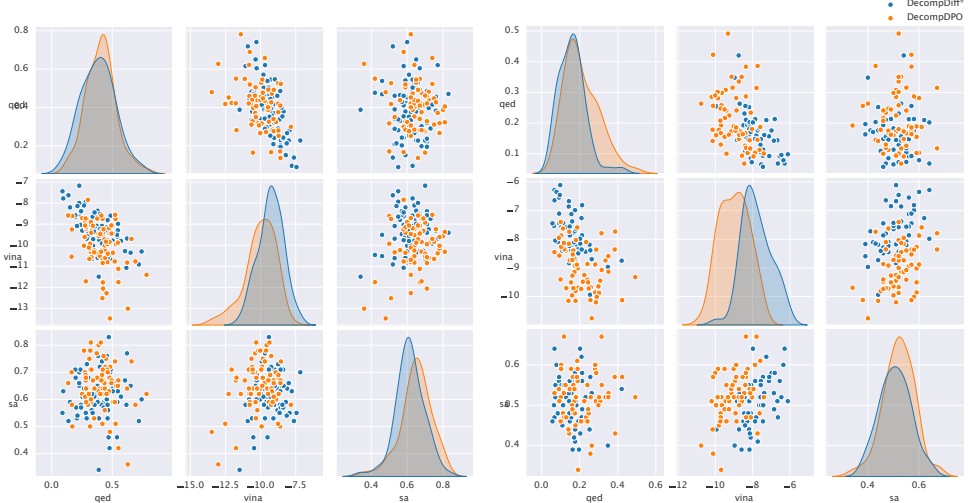

Figure 13: Pairplots of molecules' properties before and after using DECOMPDPO for molecular optimization on protein 4Z2G (left) and 2HCJ (right).

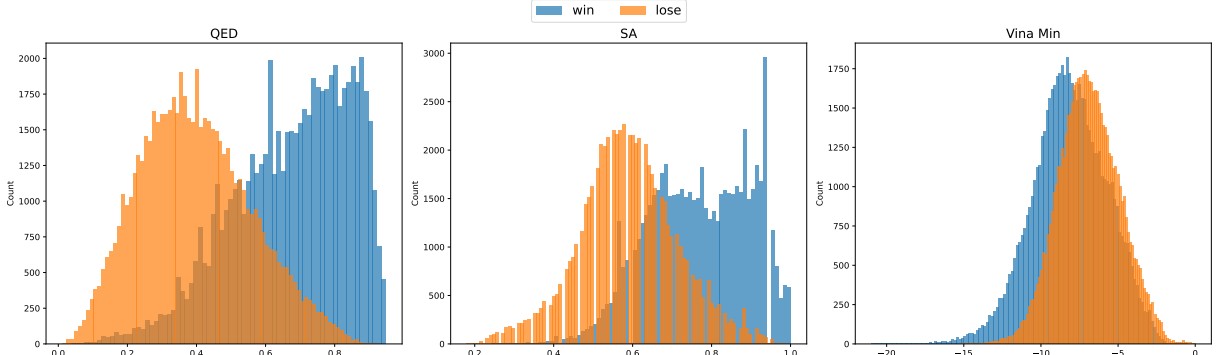

Figure 14: Distribution of QED, SA, and Vina Minimize of the winning and losing molecule in the training set.

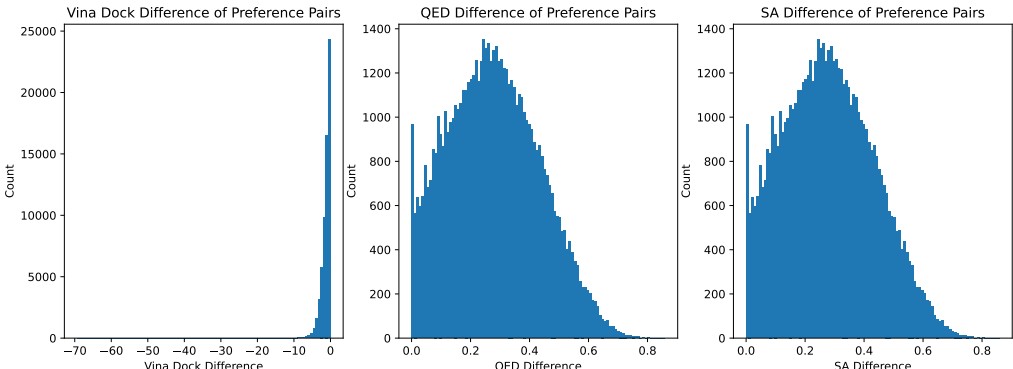

Figure 15: Distribution of difference in QED, SA, and Vina Minimize of the winning and losing molecule in the training set.

## C.6 Optimization Trend

To demonstrate that our method steadily moves toward the preferred distribution, we plot the success rate over training steps. As shown in Figure 16, the curve exhibits a clear upward trend throughout the training process.

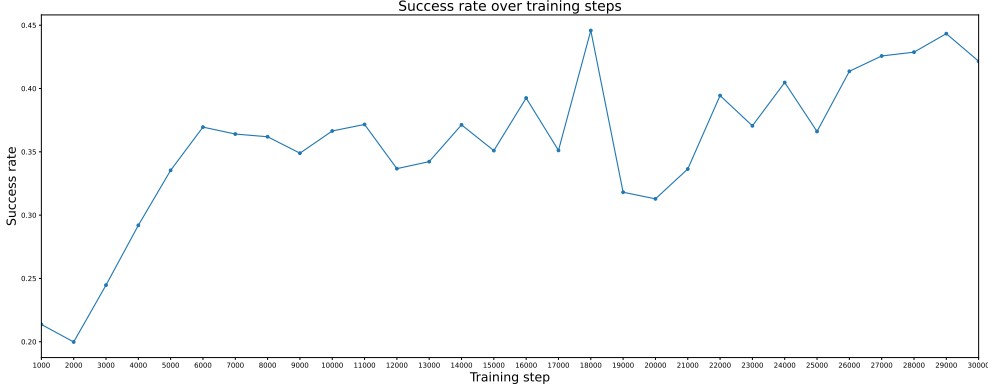

Figure 16: Success rate along the training process of DECOMPDPO on CrossDocked2020 test set.

## C.7 Examples of Generated Molecules

Examples of reference ligands and molecules generated by DecompDiff* and DECOMPDPO, which are shown in Figure 17.

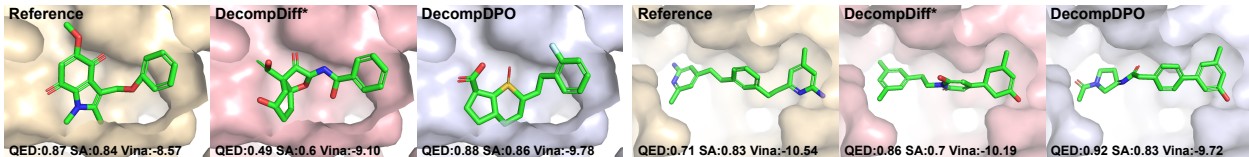

Figure 17: Additional Examples of reference binding ligands and the molecule with the highest property among all generated molecules of DECOMPDIFF* and DECOMPDPO on protein 1GG5 (left) and 3TYM (right).

