# OpenReview forum: "Decomposed Direct Preference Optimization for Structure-Based Drug Design"
_TMLR — Accepted by TMLR_

### Review · Reviewer_Vc4d · 2025-06-21

**Summary Of Contributions:**

The paper proposes DECOMPDPO, a novel structure-based drug design (SBDD) model. It introduces decomposition into optimization objectives and aligns diffusion models with pharmaceutical needs using multi-granularity preference pairs. The loss function combines GLOBALDPO and LOCALDPO terms, handling both decomposable and non-decomposable targets and providing precise control for multi-objective optimization. Additionally, DECOMPDPO uses physics-informed energy terms to penalize unrealistic molecular conformations, ensuring the generated molecules’ structural rationality.

**Audience:**

Yes

**Claims And Evidence:**

Yes

**Requested Changes:**

1. The authors should conduct a more in-depth analysis of DECOMPDPO’s theoretical properties, such as proving its convergence and stability during optimization. This will enhance the paper;s theoretical depth and provide stronger support for other researchers, promoting methodological development in the field.
2. It’s advisable to include more comparisons with the latest methods, particularly in handling complex drug design tasks. A more detailed analysis of the experimental results would also be beneficial, such as discussing performance changes under different target protein families or molecular attribute weights. This will demonstrate DECOMPDPO’s robustness and adaptability under various conditions.
3. The paper lacks a detailed evaluation of DECOMPDPO’s computational efficiency. Given the high computational cost of training and sampling in diffusion models, DECOMPDPO’s extra optimization steps may further increase time and computation costs. The authors should explore optimization strategies, such as model compression and accelerated sampling procedure, to enhance the method’s feasibility and efficiency in practical applications.
4. Although high-quality data scarcity is a bottleneck for generative models in SBDD, the paper doesn’t adequately discuss other challenges DECOMPDPO may face in practical applications, like protein structure dynamics and differences between targets. Moreover, it lacks detailed discussions on integrating DECOMPDPO into the entire development process and the practical problems and solutions that may arise.
5. The paper notes that in multi-objective optimization tasks, different targets can interfere with each other, leading to suboptimal results. DECOMPDPO uses a combination of LOCALDPO and GLOBALDPO to address this issue. However, there seem to be no ablation experiments to validate the combined loss function’s effectiveness.
6. The paper contains a few grammatical errors. Please refine them to improve readability.

**Strengths And Weaknesses:**

The experimental results on the CrossDocked2020 dataset show DECOMPDPO’s superiority in molecule generation and optimization tasks. The paper is well-structured, addresses a clear problem, employs advanced generation techniques, indicating it meets the basic requirements for publication. However, the authors need to address some issues before considering further decision.

---

> ### Author Response · Authors · 2025-07-31
>
> We thank the reviewer for time spent on our work and for the thoughtful comments. Below we address every point in the order raised and indicate the changes in the revised manuscript.
>
> **Q1: Convergence and stability during optimization**
>
> A1: We appreciate the reviewer’s request for a theoretical convergence analysis. However, for DPO or RLHF algorithms like PPO, convergence is typically not guaranteed in the general case due to non-convexity of the optimization landscape and distribution shift during fine-tuning. Instead, to demonstrate our methods steadily moving towards preferred distribution through the optimization process, we draw the curve of Success Rate over training step as shown in Appendix C.6.
>
> **Q2: More comparisons with the latest methods. More detailed analysis, such as performance changes under different target protein families or molecular attribute weights.**
>
> A2: We already benchmark DecompDPO against the latest structure-based design and optimization methods, including IPDiff, MolCRAFT, TAGMol, KGDiff, AliDiff and DecompOpt, which we believe are representative of the current state of the art performance.
> The CrossDocked2020 test set includes diverse protein families such as kinases, proteases, and nuclear-receptors. Table 2 reports results per target, demonstrating DecompDPO’s robustness across different protein families.
> To illustrate controllability we sweep the preference weights from single-objective settings to a balanced multi-objective setting and summarized the results as below. The trade-off table shows DecompDPO can efficiently balance optimization objectives as the weights change.
>
> |                     | QED |        | SA   |        | Vina Min |        |
> | ------------------- | ------- | ------ | ---- | ------ | -------- | ------ |
> |                     | mean    | median | mean | median | mean     | median |
> | DecompDiff          | 0.45    | 0.43   | 0.6  | 0.6    | \-7.6    | \-7.88 |
> | GlobalDPO - QED     | **0.48**    | **0.47**   | 0.65 | 0.65   | \-8.05   | \-8.27 |
> | GlobalDPO - SA      | 0.44    | 0.4    | **0.68** | **0.69**   | \-7.88   | \-8.02 |
> | LocalDPO - Vina Min | 0.46    | 0.43   | 0.67 | 0.67   | **\-8.97**   | **\-9.15** |
> | DecompDPO           | 0.48    | 0.46   | 0.67 | 0.67   | \-8.3    | \-8.57 |
>
> **Q3: Computational efficiency**
>
> A3: DecompDPO training for 30,000 steps takes ~10 hours with a single NVIDIA A40 GPU. We appreciate the reviewer's suggestion and will explore engineering optimizations in future development.
>
> **Q4: Practical deployment considerations**
>
> A4: The model fine-tuned with DecompDPO can be plugged directly into existing structure-based pipelines as a generator. Incorporating receptor flexibility or lead-optimisation feedback loops is beyond the present scope but remains an exciting direction for future research.
>
> **Q5: Ablation of LOCAL and GLOBAL loss**
>
> A5: As we metioned in Q2, DecompDPO excels at single-objective optimisation and achieves a balanced trade-off in the multi-objective setting, validating the effectiveness of the combined loss function.
>
> **Q6: Language and presentation**
>
> A6. We thank the reviewer for pointing this out. We have proof-read the manuscript and fixed all grammatical issues that we found.

---

### Review · Reviewer_9zyk · 2025-07-12

**Summary Of Contributions:**

The authors propose a combination of techniques for fine-tuning a diffusion-based 3D molecular generative model so it docks to a known protein pocket. The primary method selected is Direct Preference Optimization (DPO), originally developed for language models and valued for its simplicity—no explicit, complex reward design is required—and its data efficiency.

The authors realize that aligning the model with DPO only at the global molecule level may be insufficient to incentivize the generation of high-quality substructures, which they have shown contribute to better overall scores. Consequently, they also compute the DPO objective at the substructure level. This modification is straightforward: they simply sum the DPO objectives computed for each substructure (the same objective applied to smaller sets of atoms and bonds). They argue that this provides finer control over generation and leads to improved results.

Analogous to annealing the noise schedule in diffusion models to obtain higher-quality samples, the authors further propose annealing the weight of the KL divergence between the reference model and the optimized model to ensure training stability.

**Audience:**

Yes

**Broader Impact Concerns:**

The broader impact provided by the authors are more like advertising their contribution and potential of their methods again. The real impact part relating to responsible use of this method is vague.

**Claims And Evidence:**

No

**Requested Changes:**

Please see my comments above

**Strengths And Weaknesses:**

Strengths:
1. Strong performance as compared to a number of existing SOTA methods
2. Simple yet sensible methods: align DPO at both global and local structure level

Weaknesses:
1. Relatively weak contribution: though technically sound, the main contribution seems to be just adding an additional local level DPO alignment. This weakness does not affect my decision as TMLR does not focus on novelty, but rather technical correctness
2.  There are several technical inaccuracies or confusions that need to be clearly articulated

    a. The authors claim on page 5, just before the definition of Equation (5), that "$p_{theta}(M_0|P)$ in tractable for diffusion models" yet all the subsequent equations all still use the same $p_{theta}(M_0|P)$ and to make things worse jointly modelling over all time steps as $p_{theta}(M_{1:T}|P)$. There is no clear explanation of why this works, it seems non-sensical

    b. In their physically constrained optimization, the authors penalize any generated structure whose bond lengths or bond angles deviate markedly from the mean values observed in the training set. This approach seems questionable. Bond lengths and angles between adjacent atoms can vary widely for several well-known reasons—conformational isomerism, hybridization differences, resonance/delocalization, steric and torsional strain, and so on. It is therefore unreasonable to assume a single, symmetric (e.g., Gaussian) distribution for these geometric parameters. Deviation from the global mean of a multimodal distribution does not necessarily indicate an abnormal structure; the value may lie near a different local mode. Although this penalty apparently improves results, its theoretical basis is unclear. I would like to see histograms of bond lengths and angles for a representative set of substructures from the full dataset. If these distributions are indeed multimodal, the current formulation is not well-justified.

3. One of the claims of using DPO is its data efficiency to address data scarcity issue in medicinal chemistry. Author have repeatedly mentioned this in the first few section of the introduction. However, there isn't a single experiment convincingly demonstrating this. I would like to see how efficient DPO is to fully justify it usage.

4. Section 3.4 for linear beta schedule seems non-sensical. The exact words used by author are " we propose a linear beta schedule, $β_t = t/T β_T$ , where $β_T$ is the maximum regularization weight at the initial stage t = 0 and decreases linearly as t approaches T." This is kind of absurd. According to the schedule, $β_t$ actually increases overtime as t increases from 0 to T. How does this work? Furthermore, $β_t$ in equations 4,5,6,7 and 8 are merely just a multiplication factor over the entire objective in log. How does it control deviation from reference? Authors need to clearly define these terms and their rationale rigorously.

 4. There are few spelling and grammatical errors throughout the manuscript. A careful review is warranted to correct these mistakes

---

> ### Author Response · Authors · 2025-07-31
>
> We thank the reviewer for the thoughtful and detailed review. Below we address every point in the order raised and indicate the changes in the revised manuscript.
>
> **Q1: Clarification of the optimization formula**
>
> A1: The conditional likelihood $p_{\theta}(M_0 | P)$ is intractable because it requires marginalizing over every diffusion path. Following Diffusion-DPO [1], we utilized the evidence lower bound of $logp_{\theta}$ to obtain Equation 5, the expectation of the trajectory‑level log‑likelihood ratio under $p_{\theta}(M_{1:T} | M_0, P)$. This substitution yields the tractable formulation in Equation 5, making the loss fully computable while preserving the original DPO semantics. We have refined Equation 5 for greater clarity.
>
> **Q2: Justification for physically constrained optimization**
>
> A2: We additionally plot histograms for the six most frequent bond types and bond angles and added in Appendix C.2. As the figures show, these distributions all exhibit a single dominant mode, supporting our clipped quadratic penalty centred at the empirical mean is appropriate for penalty. Empirically, we have also demonstrated that our physical-informed energy terms is beneficial for both conformation energy and optimization objectives.
>
> **Q3: Data-efficiency of DPO**
>
> A3: We do not claim intrinsic data-efficiency for DecompDPO. Rather, our point is that preference alignment steers the generator toward high-reward, pharmaceutically desirable molecules more directly than distribution learning, which only imitates the training data. Because the preference pairs are obtained from molecules generated by the model itself, DecompDPO is not constrained by the size of the original medicinal-chemistry data set and is therefore less sensitive to data-scarcity issues that often arise in drug-discovery tasks.
>
> **Q4: Concerns regarding linear beta schedule**
>
> A4: The standard objective of RLHF is expressed as Equation 3, where $\beta$ weights the KL penalty that keeps the optimised model close to the reference model. DPO re-expresses the same trade-off in closed form; the KL term remains inside the log-ratio, and its gradient remains proportional to $\beta$. Consequently,$\beta$ controls how far the generator is permitted to deviate from the reference distribution at each diffusion step.
>
> We adopt a linear schedule, so that when $t$ is large, $\beta$ is large to enforce strong regularization, keeping optimization stable and preserving the distribution learned by the base diffusion model; when $t$ is small, the denoised structures already resemble realistic ligands, we use a small $\beta$ to relaxes the KL constraint and lets the optimiser follow the preference signal more aggressively. This strategy effectively balances the influence of the reference model with the optimization goals throughout the denoising process.
>
> **Q5: Fix grammatical errors**
>
> A5: We thank the reviewer for pointing this out. We carefully proof-read and corrected all typographical and grammatical errors that we found.
>
> [1] Bram Wallace, et al. Diffusion model alignment using direct preference optimization (2023).

---

### Review · Reviewer_cUBn · 2025-07-16

**Summary Of Contributions:**

The authors propose a number of changes to 3D structure-based drug design ligand generative models and 3D structure-conditioned molecular optimisation models. These changes include combining existing ideas on decomposing the ligand into fragments with reinforcement learning preference alignment methods, as well including energy based guidance terms and an improved DPO schedule. Collectively these changes aim to improve the 3D quality of generated ligand structures, as well as aligning the samples of the model to optimise pharmaceutically relevant scoring functions such as drug-likeness, synthetic accessibility, and docking scores. The authors show that assigning global scores to decomposed substructures of the molecule and fine-tuning the model using their proposed objective can improve docking scores of generated samples. They also show that penalising high-energy structures helps their model to produce more realistic structures, and they provide an improved schedule for $\beta$, the parameter which controls the deviation from the reference distribution in DPO.

**Audience:**

Yes

**Claims And Evidence:**

No

**Requested Changes:**

- Provide a more in-depth discussion or investigation into why QED and SA scores do not seem to be improved much by introducing DPO. This is also closely related to the additional ablation studies, below.
- A more thorough discussion on related work and improved positioning of this paper's contributions (see above for more detail). Ideally, this would also include running well-known models such as reinvent on their molecular optimisation or de novo generation tasks.
- Additional ablation studies focused on each of the specific changes they are proposing to make to the model (see above for more details).
- The evaluation should show the average size of ligands sampled for all models where possible. Since vina score is highly correlated with ligand size, it is important to include these numbers to allow a reasonable comparison between methods.
- The success rate definition makes no chemical sense and should modified or removed. Firstly, a QED of 0.25 is far too low to be considered drug-like, and, secondly, each target will have a very different distribution of vina scores, even for reference binders, so it does not make sense to apply the same threshold to every protein.

## Other Clarifications

- How do you deal with molecules in the training dataset that cannot be decomposed by BRICS?
- In equation 1 p is used to define a one-step forward noising process, but then in equation 6, q is used for a multi-step noising process and $p\_{\theta}$ is used for a multistep denoising process. I think the notation here could be clarified to reduce confusion.
- The notation for H in section 3.1 is not clear. What does H actually correspond to? What does the $i$ in $\eta\_{ik}$ refer to?

**Strengths And Weaknesses:**

## Strengths

- The work is theoretically justified and well presented.
- The model produces strong results for docking score optimisation compared to a number of state-of-the-art baseline methods for 3D SBDD generation.
- Ablation studies seem to show that including decomposition DPO objectives improves the predicted binding affinity of sampled molecules.

## Weaknesses

- While docking scores are improved with the authors' method, the QED and SA do not seem to show any improvement from applying decomposed DPO. At a minimum the authors should provide a discussion on this and attempt to provide some possible explanations for why this may be the case.
- The paper could do a better job of positioning itself with existing work, especially in the introduction and related works sections. The authors state their method aligns the model with "real-world pharmaceutical requirements". Which requirements? Why are they relevant? And how is this method different from the many other existing methods for SBDD which fine tune or align to a given scoring function, eg. reinvent. The authors also say they are "one of the first to introduce preference alignment to SBDD". This is not entirely true - while DPO is not commonly used in SBDD, many methods have been proposed to align the model's samples with a scoring function or with high-scoring, preferred molecules. The related work section of the paper should also provide a more thorough discussion on these existing methods, such as reinvent, GraphGA, MARS, GEAM, Saturn, etc. Additionally, given that both decomposition into molecular fragments and DPO have both been previously proposed for SBDD, I would recommend the authors to outline the specific contributions of their method more clearly.
- The authors provide some ablations studies on their design decision but more are needed to validate their ideas. They show results for a model trained without localDPO but DecompDiff* seems to perform on par with this model and has no linear $\beta$ schedule and no DPO fine-tuning, making these results puzzling. More thorough ablation studies, removing only one feature at a time from the core DecompDPO model, are needed here. These should include examine the impact of the local and global DPO objectives individually, and the physically constrained optimisation terms.

---

> ### Author Response · Authors · 2025-07-31
>
> We thank the reviewer for the thorough and constructive review. Below we address every point in the order raised and describe the corresponding revisions.
>
> **Q1: In-depth discussion about QED and SA optimization**
>
> A1: To separate the effect of preference-alignment from model capacity, we have conducted additional single-objective optimization and summarized as below. When DPO is applied solely to QED or SA the mean values rise from 0.45 to 0.48 for QED and from 0.60 to 0.68 for SA without degrading docking affinity, which is signficantly improved from the base model. Considering DPO using generated molecules from original landscape to guide model towards preference, this performance potentially further improved with iterative DPO.
>
> |                     | QED |        | SA   |        | Vina Min |        |
> | ------------------- | ------- | ------ | ---- | ------ | -------- | ------ |
> |                     | mean    | median | mean | median | mean     | median |
> | DecompDiff          | 0.45    | 0.43   | 0.6  | 0.6    | \-7.6    | \-7.88 |
> | GlobalDPO - QED     | **0.48**    | **0.47**   | 0.65 | 0.65   | \-8.05   | \-8.27 |
> | GlobalDPO - SA      | 0.44    | 0.4    | **0.68** | **0.69**   | \-7.88   | \-8.02 |
> | LocalDPO - Vina Min | 0.46    | 0.43   | 0.67 | 0.67   | **\-8.97**   | **\-9.15** |
> | DecompDPO           | 0.48    | 0.46   | 0.67 | 0.67   | \-8.3    | \-8.57 |
>
> **Q2: A more thorough discussion on related work**
>
> A2: We appreciate the opportunity for further clarification. Our focus is structure-based drug design, where the ligand is generated conditioned on the 3D protein pocket. The methods including reinvent, GraphGA, MARS, GEAM, and Saturn cited by the reviewer generate 2D graphs or SMILES without 3D protein input. Comparison with these methods can be uninformative as ligand size, search space and evaluation metrics differ substantially, so we instead select structure based drug design or optimization method as baselines.
>
> However, thanks to the suggestion, and now we added the discussion of these methods in the Related Work section to highlight the distinction. We would also like to highlight it here that unlike reinforcement-learning schemes that learn a reward model such as reinvent, DecompDPO aligns the generator directly with preference pairs and therefore requires no extra policy network or reward predictor.
>
> **Q3: Additional ablation studies**
>
> A3: We've conducted an abalation study to validate the benefit of physically constrained optimisation terms, which is provided in Appendix C.2. For better clarity, we summarized the result here. As the result shows, physically constrained optimisation terms not only leads to substantial improvement in conformation energy related metrics, but also in optimization objectives.
>
> || QED || SA   || Vina Min ||
> | ---------- | ------- | ------ | ---- | ------ | -------- | ------ |
> || mean    | median | mean | median | mean     | median |
> | DecompDiff | 0.45    | 0.43   | 0.6  | 0.6    | \-7.6    | \-7.88 |
> | w/o phys   | 0.47    | 0.44   | 0.65 | 0.65   | \-8.07   | \-8.32 |
> | DecompDPO  | **0.48**    | **0.46**   | **0.67** | **0.67**   | **\-8.3**    | **\-8.57** |
>
> ||JSD Dist | Energy Diff - rigid fragments | Energy Diff - Mol | RMSD - rigid fragments | RMSD - Mol |
> | --------- | ------------ | ----------------------------- | ----------------- | ---------------------- | ---------- |
> | w/o phys  | 0.07         | 48| 887.15| 0.12| 1.1|
> | DecompDPO | 0.07         | **38.38**| **672.02**| **0.11**| **1.08**       |
>
> **Q4: Include average size of ligands sampled**
>
> A4: We thank the reviewer for the suggestion. Average heavy-atom counts are now reported in Table 1 and Table 4.
>
> **Q5: Definition of success rate**
>
> A5: Following [1,2], we define the threshold of Success Rate, which are chosen based on chemical insights. The Vina threshold (< -8.18 kcal/mol) corresponds to 1uM binding affinity, which is a common requirement for a potential drug candidate in practical drug discovery; the QED and SA thresholds are the 10th percentile of DrugCentral [3] (a database of up-to-date drugs and pharmaceuticals), to reflect the latest drug property distribution.
>
> **Q6: How do you deal with molecules in the training dataset that cannot be decomposed by BRICS?**
>
> A6: As noted in Appendix B.5, we simply discard molecules that cannot be decomposed or reconstructured.
>
> **Q7: Confusion about p and q**
>
> A7: We thank the reviewer for pointing this out. We renamed the forward noising distribution to $q$ and updated the equations.
>
> **Q8: clarification of H**
>
> A8: Matrix H maps atoms to priors. Row $i$ has a single 1 indicating the substructure to which atom $i$ belongs.
>
> [1] Long, Siyu, et al. Zero-Shot 3D Drug Design by Sketching and Generating. NeurIPS (2022).
>
> [2] Guan, Jiaqi, et al. DecompDiff: Diffusion Models with Decomposed Priors for Structure-Based Drug Design. ICML (2023).
>
> [3] Ursu, Oleg, et al. DrugCentral: online drug compendium. Nucleic acids research (2016).

---

> > ### Comment · Reviewer_cUBn · 2025-08-04
> >
> > Thank you for your answers and additional experiments! However, I still have some remaining concerns, as follows:
> >
> > 1. Many existing methods do use the 3D protein structure to optimise molecular generation, but they only do this indirectly by computing docking scores. If you wish to claim that you model is an important contribution to molecular optimisation, then I would still recommend you provide a comparison with some of these methods (particularly newer versions of REINVENT since it is so commonly used in practice), although I don't see it as a requirement for the purposes of this paper. Both methods aim to optimise a scoring function, in your case SA + Vina + QED, so all of these metrics still hold for both (with the caveat of being careful to take into account ligand size and optimisation efficiency). Reinvent also does not use a reward model, just a non-differentiable scoring function.
> >
> > 2. The improvement on QED and SA scores is very minor, so again I would encourage to investigate or discuss why this might be.
> >
> > 3. With the ligand sizes now shown in the results table, it's clear that DecompDPO generates larger ligands than other methods. Since this is known to have be strongly correlated with docking score I recommend you control the size of ligands that are generated, if possible, to allow a fair comparison with existing methods. Alternatively you could present results normalised by ligand size.
> >
> > 4. The success rate metric still makes no chemical sense. You are correct that you can convert a docking energy to a binding affinity (for a single protein in an experimental setup), but docking score has very little correlation with docking energy. Hence the goal for computational optimisation should still be to optimise the docking score, but your current definition will be heavily biased by whichever proteins have naturally lower docking scores. Additionally, a QED threshold of 0.25 will also catch many undesirable molecules which would never actually be synthesised, meaning it's questionable what the value of this threshold is.
> >
> > Overall, I think the paper investigates some interesting questions, but currently the results seem somewhat misleading - minor improvements in SA and QED, docking score improvements that may only be due to ligand size increases, and a questionable success rate metric. Personally, I have no problem if you publish negative results, but the results must be fair. Additionally, I think the paper could do a better job of positioning itself - why is it useful to combine DPO with ligand decomposition, which have both been previously applied to the same problem? What does this achieve (or even just aim to achieve) that others don't?

---

> > > ### Author Response · Authors · 2025-08-15
> > >
> > > We thank the reviewer for the thoughtful, constructive feedback. We answer every point in the order raised as below.
> > >
> > > **Q1: Should we compare with REINVENT-style optimizers?**
> > >
> > > A1: Thank you for the thoughtful suggestion. We fully agree that docking-in-the-loop optimizers (e.g., REINVENT) are widely used and valuable in practice. For this manuscript, however, our goal is to evaluate structure-conditioned 3D SBDD generation, and we therefore keep comparisons within that paradigm for the following reasons:
> > >
> > > - Different problem formulations. SBDD (including our model and optimization baselines like TAGMol, KGDiff, AliDiff, RGA, DecompOpt) directly **condition on the 3D pocket and generate 3D poses** under geometric and steric constraints during generation. LBDD optimization methods (including REINVENT-style methods) optimize **2D graphs/SMILES** using property/docking scores **without explicit 3D protein conditioning** at generation time. Mixing these settings confounds inputs and hypothesis spaces, making 'wins/losses' hard to interpret.
> > >
> > > - Fairness and controls. Although metrics like QED/SA/docking score overlap, the feasible set and compute budget differ substantially:
> > >
> > >   - 3D SBDD methods must satisfy pose realism as they generate; 2D methods depend on post-hoc docking (choice of docking engine/version, pose count, protonation/tautomer enumeration, and docking budget), each introducing variance that can dominate reported numbers.
> > >
> > >   - Size effects are harder to normalize across paradigms because 2D optimizers can freely expand search over graph space, while 3D methods are implicitly constrained by pocket geometry.
> > >
> > > - Field convention. Recent SBDD works benchmark primarily within 3D structure-conditioned families, precisely to avoid the above confounders. We follow this practice to provide the most apples-to-apples assessment of structure-aware optimization.
> > >
> > > That said, we appreciate the reviewer's interest in a bridge to ligand-only optimizers. In the revision we have made the scope distinction explicit in the Related Work and explained that our main baselines are drawn from the former.
> > >
> > > We hope this clarifies our rationale: we deliberately benchmark within the structure-conditioned SBDD paradigm to ensure a fair, interpretable comparison, while acknowledging REINVENT-style methods as a complementary line of work.
> > >
> > > **Q2: Discussion about QED and SA improvement**
> > >
> > > A2: Two factors potentially affect QED and SA optimization results:
> > >
> > > - Pair-selection bias from the weighted score. When constructing preference pairs for multi-objective training, we use a weighted sum over Vina, QED, and SA. In our data, the win–lose deltas have different dynamic ranges: as shown in Figure 15, ΔVina exhibits a long negative tail, whereas ΔQED and ΔSA are much narrower, indicating Vina potentially influences which pairs enter training and the magnitude of DPO updates more, so QED/SA potentially receives a weaker effective signal.
> > >
> > > - 3D–2D metric mismatch. The model edits 3D pocket-constrained poses and fragment placements, while QED and SA are computed purely from 2D molecular topology. Preference pairs that differ in pose can yield substantial docking gains with atom position loss yet provides little informative supervision for QED and SA within DPO.
> > >
> > > We leave remedies such as exploring remedies such as adjusting weight for pair construction, modifying the DPO objective to better adjust gradients across properties, and using iterative DPO to future work.
> > >
> > > **Q3: Concerns regarding ligand size**
> > >
> > > A3: To further illustrate the effect of ligand size, we re-evaluated with ref prior and beta prior. In both settings, heavy-atom count remained essentially unchanged, while DecompDPO consistently outperformed DecompDiff across all metrics. These improvements therefore cannot be simply attributed to ligand-size increase, but to stronger generative model.
> > >
> > > ||Vina Score||Vina Min||Vina Dock||High Affinity||QED||SA||Diversity||Size|Success Rate|
> > > |-|-|-|-|-|-|-|-|-|-|-|-|-|-|-|-|-|
> > > ||mean|median|mean|median|mean|median|mean|median|mean|median|mean|median|mean|median|mean|mean|
> > > |DecompDiff-ref prior|\-5.78|\-5.84 |\-6.48|\-6.41 |\-7.37|\-7.46|56.90%|59.20%|0.51|0.52|0.67|0.66|0.69|0.7|21.7|21.10%|
> > > |DecompDPO-ref prior|\-6.77|\-6.54 |\-7.38| \-7.08 |\-8.25|\-8.22|84.20%|94.90%|0.57|0.57|0.73|0.74|0.71|0.74|21.47|37.90%|
> > > |DecompDiff-beta prior|\-5.67|\-7.1|\-7.6|\-8.15|\-8.94|\-9.14|76.00%|94.16%|0.39|0.36|0.58|0.59|0.57|0.57|34.29|21.84%|
> > > |DecompDPO-beta prior|\-6.35|\-8.23|\-8.97|\-9.7|\-10.66|\-10.69|90.39%|99.45%|0.42|0.38|0.66|0.67|0.64|0.64|34.37|39.00%|
> > >
> > > **Q4: On the Success Rate metric**
> > >
> > > A4: We understand the reviewer's concern about the chemical interpretability of Success Rate. We retain Success Rate as a practical, single-number summary of multi-objective quality, and report it alongside absolute metric values and mean heavy-atom counts in the main table. This makes the optimization results and the effect of molecule size transparent.

---

### Decision · Action_Editor_q8T5 · 2025-08-25

**Recommendation:** Accept as is

**Audience:**

Yes

**Audience Explanation:**

The field of SBDD currently actively researched, and has arguably not yet converged. Insights and improvements are surely interesting to the community, even if minor or incremental.

All reviewers are positive on audience criterion.

**Claims And Evidence:**

Yes

**Claims Explanation:**

The paper claims to improve structure-based drug design diffusion models by scheduling tweaks, energy-based rewards, and preferential optimisation. These claims are relatively modest, and the improvement is especially on vina scores at the expense of QED. Two reviewers agree on the claims and evidence, and one reviewer raises concerns about misleading or confusing results. The authors have engaged in useful discussion.

Overall the paper’s experiments are relatively extensive, include ablations, and show largely positive and mostly consistent results. The paper’s claims are supported by sufficiently clear and convincing evidence.